



# Quantifying lahar damage using numerical modelling

Stuart R. Mead[1,2], Christina Magill[1], Vincent Lemiale[2], Jean-Claude Thouret[3] and Mahesh Prakash[2]

[1]Risk Frontiers, Department of Environmental Science, Macquarie University, Sydney, Australia
[2]Commonwealth Scientific and Industrial Research Organisation, Clayton 3168, Victoria, Australia
[3]Laboratoire Magmas et Volcans UMR6524 CNRS, IRD and OPGC, University Blaise Pascal, Campus Les Cézeaux, 63178 Aubière , France

*Correspondence to:* Stuart R. Mead (Stuart.Mead@mq.edu.au)

**Abstract.** Lahars are volcanic flows containing a mixture of fluid and sediment that have caused significant
damage to buildings, critical infrastructure and human life. The extent of this damage is controlled by properties of the lahar, location of elements at risk and susceptibility of these elements to the lahar. Here we focus on understanding lahar-induced building damage. Quantification of building damage can be difficult due to the complexity of lahar behaviour (*hazard*), uncertainty in number and type of buildings exposed to the lahar (*exposure*) and the uncertain susceptibility of buildings to lahar induced damage (*vulnerability*). In this paper, we
quantify and examine the relative importance of lahar hazard, exposure and vulnerability in determining building damage with reference to a case study in the city of Arequipa, Peru. Numerical modelling is used to investigate lahar properties important in determining the inundation area and forces applied to buildings. Building vulnerability is quantified through the development of critical depth–pressure curves based on the ultimate bending moment of masonry structures. In the case study area, results suggest that building strength plays a minor
role in determining overall building losses in comparison to the effects of building exposure and lahar hazard properties such as hydraulic characteristics of the flow.

**Keywords** lahar · hazard · building vulnerability · rheology · simulation

## Introduction

Lahars, defined as gravity-driven flows containing a mixture of volcanic sediment and water (Vallance and
Iverson, 2015), have caused severe damage to infrastructure and buildings (e.g. de Bélizal et al., 2013; Pierson et al., 2013; Ettinger et al., 2015; Jenkins et al., 2015) in addition to being responsible for a large proportion of volcanic fatalities (Auker et al., 2013). Assessing the extent of potential lahar damage can be difficult due to the complexity of flow behaviour, uncertainty in the number of elements exposed to lahars (e.g. buildings and bridges) and a lack of knowledge in the structural capacity of these elements to withstand damage causing components of
the lahar flow. Using the common definitions of Varnes (1984), we define the damaging components of lahar flow (e.g. velocity, depth and density) as the hazard; environmental characteristics of exposed elements (e.g. building location and orientation) as the exposure; and the ability of exposed elements to withstand the hazard (e.g. building strength) as vulnerability. Lahar induced damage is controlled by the interactions between these factors; however, the relative importance of each component can vary. Here we focus on quantifying and examining the relative
importance of hazard, exposure and vulnerability in determining lahar induced building damage.

Post-event field assessments of building damage can elicit information relating lahar hazard to structural damage. However, these field assessments tend to only record information on substantial damage, are affected by terrain




changes during the event which alters exposure and often rely on a-priori assumptions of building strength and vulnerability (Ettinger et al., 2015). Pre-event assessments are affected by the lack of reliable hazard intensity

measures (van Westen et al., 2006; Ettinger et al., 2015), differences in spatial and temporal scales, uncertainty surrounding site-specific lahar triggers (Di Baldassarre and Montanari, 2009) and a lack of structural information on building stock (Ettinger et al., 2015). These issues are reflected in the relative lack of studies on hazard impact in urban areas (Jenkins et al., 2015) and often results in a reliance on expert judgement to develop vulnerability models for lahars and flash floods (Ettinger et al., 2015).

The physical vulnerability of buildings, defined as the susceptibly of a building to damage with respect to the hazard (Künzler et al., 2012), is a function of building characteristics such as size, shape, age, construction materials, structural integrity, maintenance and build quality (Martelli, 2011; Künzler et al., 2012; Ettinger et al., 2015). Information on these building properties is often lacking and hard to collect on a large scale. This often leads to the simplification of vulnerability into a measure that can provide a relative indication of vulnerability

and consequent damage (Künzler et al., 2012). Studies simplifying vulnerability into a relative index use a combination of qualitative and quantitative metrics obtained through building surveys, interpretation of remote sensing data and GIS techniques to map and analyse vulnerability on a large scale (e.g. Lavigne, 1999; Künzler et al., 2012; Galderisi et al., 2013; Thouret et al., 2013; Thouret et al., 2014; Ettinger et al., 2015). These methods can be applied to understand and highlight spatial patterns in vulnerability; however, as a relative measure, they

cannot provide guidance on expected damage for any specific event.

A direct estimation of damage caused by specific events requires quantified relationships describing a buildings response to the hazard. Buildings can be damaged through a number of mechanisms including: (i) direct damage resulting from static and dynamic forces imposed by the flow; (ii) damage to foundations through erosion and scour; (iii) buoyancy effects of the flow causing structures to float; (iv) direct damage from larger debris (missiles)

within the flow; and (v) indirect damage caused by chemical and biological actions such as seeping induced weakness of mortar (Kelman and Spence, 2004). All these actions, with the exception of chemical and biological effects, are related directly to lahar depth, velocity or a combination of depth and velocity. As a result, a common approach in determining building damage thresholds for a particular building type is to relate damage to hazard intensity measures of depth and/or velocity (e.g. Zanchetta et al., 2004; Custer and Nishijima, 2015; Jenkins et

al., 2015). However, building typologies are affected by socio-economic, cultural and institutional conditions (Künzler et al., 2012) and environmental factors such as local elevation, distance from main channels and orientation affect flow depths and velocities near buildings (Thouret et al., 2014). These issues cause direct vulnerability relationships to be site-specific and requires detailed investigation of regions at risk to examine the relative importance of hazard, exposure and vulnerability on building loss.

We attempt to quantify and examine the components that determine building damage in a small area within the city of Arequipa, Peru. A relative index of vulnerability on a city-block scale was developed for Arequipa in Thouret et al. (2013) and Thouret et al. (2014). The studies by Thouret et al. highlighted two groups of vulnerability indicators, building characteristics and the physical setting, that play an important role in determining vulnerability within Arequipa. In this study, we separate the building characteristics from physical setting (i.e.

exposure) in order to identify strategies that may reduce building loss within Arequipa. Physical vulnerability of buildings is explicitly separated from exposure through the development of a building damage model dependent on flow velocity, depth and sediment concentration. Simulations of lahar flow using smoothed particle



hydrodynamics are used to examine how flow characteristics and the physical setting of city blocks affects forces on buildings and the consequent damage. While, for reasons explained earlier, damage functions presented here

are necessarily specific to Arequipa, the hazard modelling approach and vulnerability model development are described in detail to support risk assessment in other regions affected by lahars.

**Case study: Quebrada Dahlia, Arequipa, Peru**

The central business district of Arequipa, the second largest city in Peru, is situated 17 km southwest of the summit of El Misti (Fig. 1), a steep stratovolcano with a history of explosive eruptions. Rapid population growth since

1960 has resulted in an expansion of the city towards the ring plain and steep slopes of El Misti (Thouret et al., 2013). Arequipa is drained by several ravines (locally called quebradas or torrenterras), shown in Fig. 1, that have been shaped by lahars and floods originating from the volcano on volcanoclastic fans northeast of the city. These quebradas are normally dry but carry water sporadically during the December to March rainy season (Vargas Franco et al., 2010; Martelli, 2011; Thouret et al., 2013; Sandri et al., 2014). Flash floods and hyper-concentrated

flows occur relatively frequently in the quebradas, with return periods between 2 and 10 years (Vargas Franco et al., 2010; Thouret et al., 2013). Previous studies of lahar hazard and vulnerability for Arequipa identified seven alluvial terraces (T0, T1, T1', T2, T2', T3 and T4) based on stratigraphy and local elevation above the quebrada and the Rio Chili valley (Thouret et al., 2013; Thouret et al., 2014). The likelihood of inundation by a lahar or flash flood decreases with each terrace. Terrace levels T0 and T1 (up to 3m above the quebrada) are frequently

flooded (i.e. 2-10 years). The higher terraces (T1' to T2', 3 to 10 m above the quebrada) are rarely flooded (estimated 20-100 years) and the highest terraces (T3 and T4) are only likely to be inundated by lahars linked to large eruptions (Thouret et al., 2013; Thouret et al., 2014). A city wide vulnerability study by Thouret et al. (2014) identified that the city blocks most vulnerable to flash floods and lahars were on lower terraces and typically within 100 metres of a quebrada.

To build on this study and investigate the vulnerability of the quebrada channel and banks in detail, simultaneous photogrammetry and building surveys were undertaken along short sections (approximately 200 m) of several quebradas during September 2013. Here we focus on one 150 m long section of Quebrada Dahlia to examine lahar hazard and building damage. Quebrada Dahlia is a small tributary of Quebrada Mariano Melgar-Huarangal (Fig. 1), which is situated in the Mariano Melgar District on the north-easternmost fan of Arequipa, shown in detail in

Fig. 2. The case study area was chosen for the following reasons:

- The quebrada channel is relatively straight, reducing the effect of bends in the watercourse on lahar dynamics.
- Building quality varies from well-built reinforced masonry buildings to makeshift structures with little to no mortar. This allows for an investigation of the relative effects of building quality on damage caused

110        by lahars.
- All buildings are situated on the lowest terraces (T0 – T1', 1 - 5 m above the channel), meaning they may be affected by even the smallest events identified in Vargas Franco et al. (2010) and Thouret et al. (2013).

A three-dimensional reconstruction of the terrain and buildings along Quebrada Dahlia was created using the photogrammetry method described in Mead et al. (2015). The generated terrain model contained 1.4 million points

with a surface density of between 150 and 750 points per $m^2$, a GNSS-D survey undertaken in October 2014





enabled geo-referencing of the terrain reconstruction for possible inclusion in future GIS applications. The reconstruction, shown in Fig. 2c, contains 23 surveyed buildings. These buildings are separated into five groups, referred to hereafter as 'blocks', by cross streets and the quebrada. Building surveys were undertaken in 2013 and used the general approach of Thouret et al. (2014). Buildings were separated into 8 structural types (1A – 8C)
depending on construction material, roof type and structural support. These types were then grouped in larger groups of vulnerability classes. Using this building classification system, the study area contains 8 type-*A0* buildings, 7 type-*A* buildings and 8 type-*B* buildings (see Table 1).

**Developing building vulnerability relationships**

Buildings and infrastructure can be damaged through a variety of mechanisms brought upon by actions of a lahar.
Here, as in most other studies of lahar damage (Zanchetta et al., 2004; Toyos et al., 2008; Ettinger et al., 2015; Jenkins et al., 2015), we focus on the direct damage resulting from hydrostatic and hydrodynamic forces applied to buildings. We regard these actions as the most important, although scour and large debris missiles within the flow can also cause significant damage (Jenkins et al., 2015). Scour and debris actions are neglected here as they are currently too difficult to predict and incorporate into large scale loss analyses (Kelman and Spence, 2004),
particularly in regions with limited hazard and exposure information.

The building stock within Arequipa is characterised mostly by masonry structures of varying quality, with some reinforced concrete structures (Thouret et al., 2014). Therefore, we develop vulnerability relationships that are primarily focused on masonry buildings. A structural failure model similar to those employed by Roos (2003), Custer and Nishijima (2015) and Zeng et al. (2015) is implemented. Stresses for buildings in Arequipa are
calculated using the approach specified in Australian Standard (AS) 3700-2011. While some specifications in the standard may not be relevant for Arequipa, the calculation method is still valid for the area provided construction material properties from Arequipa are used as inputs. In these models, masonry walls are presumed to fail when the applied bending moment and shear forces are greater than the calculated ultimate bending moment and shear force the walls can withstand. We only consider the maximum bending moment here as preliminary investigations
suggested the force required to overcome the ultimate moment was consistently lower than the force required to overcome the ultimate shear force. The ultimate bending moment ($M_u$) is calculated using the following equation (Roos, 2003):

$$M_u = (f_t + f_d)\frac{wb^2}{6} \tag{1}$$

where $f_t$ is the tensile strength of the masonry wall, $f_d$ is the design compressive stress acting on the wall, $w$ is the
width of the wall facing the flow and $b$ is the width of bricks used in the wall. The tensile strength is assumed to be 0.2 MPa as, according to AS3700-2011, the tensile strength should be no greater than this value. The brick width, $b$, is between 150 and 250 mm for terracotta bricks (Martelli, 2011) and is assumed to be similar for ignimbrite bricks observed in the study area. The design compressive stress, $f_d$, can be determined by calculating the normal forces (i.e. building weight) acting on the walls. This can be estimated from building properties such
as number and weight of floors, weight of the masonry and building design (e.g. Roos, 2003). However, such detailed building data is lacking and carries considerable uncertainty for a heterogeneous urban area with varied construction materials, building ages and designs such as Arequipa. Instead we use the design compressive capacity ($f_o$), specified in AS3700-2011, to determine the design compressive stress:


$$f_o = \phi f_c A_b \tag{2}$$

$$f_d = k f_o \tag{3}$$

where $f_c$ is the characteristic compressive strength of the masonry, $\phi$ is the capacity reduction factor, $A_b$ is the bedded area of the masonry (brick width × length) and $k$ is a reduction factor based on the wall design. The characteristic compressive strength is determined using the unconfined compressive strength tests of Martelli (2011) on building materials sourced from Arequipa. Presuming the mortar is of relatively low quality (M2), the

characteristic compressive strengths (according to AS3700-2011) are 3.8 MPa for ignimbrite masonry and between 3.5 and 4.54 MPa for terracotta masonry. The slenderness reduction factor, $k$, describes the susceptibility to buckling. Following AS3700-2011, this factor is calculated as

$$k = 0.67 - 0.02(S_{rs} - 14) \tag{4}$$

for buildings with a reinforced concrete roof or floor (i.e. typologies 3-6C), and

$$k = 0.67 - 0.025(S_{rs} - 10) \tag{5}$$

for buildings with other roof or floor supports (typologies 1-2). This factor requires a calculation of the slenderness ratio, $S_{rs}$:

$$S_{rs} = \frac{a_v H}{k_t b} \tag{6}$$

where $H$ is the height between floors or supports, taken as 2.8 metres for reinforced concrete type buildings and

3 metres for non-reinforced buildings (Martelli, 2011). The vertical slenderness coefficient, $a_v$, is determined from the lateral support along the top edge of the wall. Walls with roof support (types 3-6C) have a coefficient of 1, while unsupported walls (types 1A-2B) act as a cantilever and have a coefficient of 2.5. Considering the thinnest bricks, the slenderness coefficient is negative for building types 1A-2B as the design is out of the range of those considered in AS3700-2011. Acknowledging the low strength of these frequently makeshift structures, the

slenderness coefficient is therefore set to 0.01. The thickness coefficient, $k_t$, takes into account the strength of supporting columns. This coefficient is set to 1 for non-reinforced frame buildings and is dependent on the spacing and thickness of reinforced beams within the masonry for reinforced buildings. Estimates of the spacing and thickness suggest that the coefficient will be between 1.4 and 2 for type 6A-6C buildings and between 1 and 1.2 for type 4 buildings. The large spacing between reinforced columns and their relative width, pictured in Thouret

et al. (2014), are responsible for the much lower coefficients assigned to type 4 buildings.

**Critical depth-pressure curves**

The range of design compressive stress for each building typology is shown in Fig. 3. The range was obtained by calculating the design compressive stress for every configuration of masonry compressive strength ($f_c$), brick width ($b$) and thickness coefficient ($k_t$). Buildings with reinforced frames (types 3, 4 and 6) are able to withstand

much greater compressive stresses than non-reinforced buildings (types 1, 2, 5). The brick width has a large effect on building strength, which is consistent with observations of Jenkins et al. (2015). Notably, the design compressive stresses are similar for building types that share the same vulnerability class identified in Thouret et al. (2014), based on the vulnerability classes of Zuccaro et al. (2008). Given these similarities, we also use the vulnerability classes (A0, types 1A-2B; A, types 3 and 5; and B, types 4 and 6A-6C) from Thouret et al. (2014).

The critical height and dynamic pressure required to overcome the ultimate bending moment (equation 1) for each vulnerability class is shown in Fig. 4. The curves in Fig. 4 indicate the point at which the applied moment from hydrostatic and dynamic pressure equals the ultimate moment of the wall. Combinations of depth and pressure



that fall above the curves indicate an applied moment greater than the building can withstand. Conversely, combinations of depth and pressure that fall below the curves indicate an applied moment less than the maximum the building can withstand. Figure 4 shows that the critical depth decreases with density of the flows as the hydrostatic pressure gradient is much larger for sediment-rich lahars. The critical depths and pressures are also affected by the vulnerability class, with *A0* structures being much less resilient than *A* and *B* structures. However, brick width has the most dominant effect on determining the strength of buildings. Wider bricks increase the section modulus ($wb^2/6$ in equation 1), resulting in stiffer walls that also have a higher compressive stress capacity.

**Lahar numerical modelling and results**

**Lahar rheology**

The bulk flow behaviour of lahar sediment-water mixtures is controlled by the relative concentration of sediment within the flow (Dumaisnil et al., 2010). In particular, the clay content and proportion of fine sediment in suspension will greatly influence the transition from a Newtonian (i.e. constant viscosity and zero shear strength) to non-Newtonian fluid (Pierson, 2005). This transition and lahar flow behaviour is affected by inter-particle interactions (collisions and electrochemical attractions), particle-bed interactions and particle-fluid interactions, the combination of which is complex and likely exists as a continuous process (Pierson, 2005). Flows can be predominantly Newtonian with sediment volume concentrations of up to 35%, provided there are few clay or fine particles present. The inter-particle interactions between larger proportions of fines or coarse sediment in the fluid will result in a small but measurable yield strength. Flows with a non-zero yield strength are often termed hyperconcentrated (Pierson, 2005; Manville et al., 2013) and can also be characterised by a marked dampening of turbulence (Pierson and Costa, 1987). At higher sediment concentrations, particle-particle collisions and the internal friction between particles begin to dominate, causing the yield strength to increase significantly. These flows tend to exhibit commonly observed lahar behaviours such as suspension of large boulders, unsorted particle deposits and rapid consolidation of the deposit as the pore fluid drains. Viscosity of the pore fluid also plays an important role in moderating the effect of inter-particle interactions (Doyle et al., 2010). At low viscosities (e.g. water), inertial forces and particle collisions dominate energy transfer within the fluid, while at higher viscosities (e.g. with a significant proportion of clay) the energy is mostly dissipated through fluid-particle interactions.

Several single-phase rheological models have been proposed to describe non-Newtonian lahar behaviour; most of these models follow the general form of the Herschel-Bulkley equation (Manville et al., 2013):

$$\tau = \tau_y + \mu\dot{\gamma}^m \tag{7}$$

where $\tau$ is the shear stress, $\tau_y$ the yield strength, $\mu$ the viscosity, $\dot{\gamma}$ the shear rate, and $m$ the shear power, which describes the response to shear (i.e. shear-thickening or shear-thinning). A simple rheological description for lahar flows assumes they behave as a viscoplastic material, commonly called a Bingham fluid. Bingham fluids have a non-zero shear strength and shear power, $m$, of 1. Flows of Bingham fluids typically have two components consisting of a basal shearing layer (shear layer) topped with a non-shearing plug layer (Rodriguez-Paz and Bonet, 2004). In more general terms, O'Brien et al. (1993) and Jan and Shen (1997) describe the total shear stress of generic sediment-water flows as being controlled by the summation of all the debris flow strength components: cohesive yield strength, Mohr-Coulomb shear stress, viscous shear stress, turbulent shear stress and the dispersive (particle collision) shear stress. At high viscosities and relatively low velocities, the turbulent stresses can be



assumed as negligible (Pierson and Costa, 1987; O'Brien et al., 1993; Jan and Shen, 1997). By combining all the relevant stresses, a generalised quadratic rheology model (Julien and Lan, 1991; O'Brien et al., 1993; Jan and Shen, 1997) can be expressed as:

$$\tau = \tau_y + \mu\dot{\gamma} + \alpha\dot{\gamma}^2 \tag{8}$$

where $\alpha$ is the turbulent-dispersive parameter, a coefficient that combines the effects of turbulence and dispersive stresses caused by sediment collisions. Rheological parameters for $\tau_y$, $\mu$ and $\alpha$ were studied for debris flows by Phillips and Davies (1991) and O'Brien and Julien (1988). We chose to implement and use this quadratic rheology model for lahars as it summarises the principal components of non-Newtonian lahar behaviour, namely a yield strength, viscous effects and a dilatant (shear-thickening) effect from particle collisions.

**Implementation in smoothed particle hydrodynamics (SPH)**

The quadratic rheology model is implemented using smoothed particle hydrodynamics (SPH) to simulate lahar flows along the case study area. SPH is a Lagrangian method that tracks the physical motion of interpolation points (commonly referred to as particles) through space. It is well suited to modelling free surface fluid flows, predicting and tracking the motion of dynamic objects within the flow (e.g. Cleary et al., 2012; Prakash et al., 2014; Cleary et al., 2015), and modelling complex flooding scenarios involving interactions with buildings (e.g. Mead et al., 2015). The SPH method used here is described in Cleary and Prakash (2004) and Prakash et al. (2014). Non-Newtonian lahar rheology was implemented in SPH using an apparent Newtonian viscosity ($\eta$). Assuming the fluid is isotropic, constitutive equations for rheology can be written as a generalised Newtonian fluid in terms of the apparent viscosity:

$$\tau = \eta\dot{\gamma} \tag{9}$$

When the apparent viscosity is constant the fluid is Newtonian with a viscosity of $\eta$. Non-Newtonian fluids can be modelled using equation 3 by developing relationships for $\eta$ based on constitutive equations (Mitsoulis, 2007). Using this approach, the apparent viscosity for the quadratic rheology is:

$$\eta = \frac{\tau_y}{\gamma} + \mu + \alpha\dot{\gamma} \tag{10}$$

Here, we also use the viscosity regularisation approach of Papanastasiou (1987), described in Mitsoulis (2007) and Minatti and Paris (2015). Regularisation is required as the apparent viscosity approaches infinity at low strain rates when using equation 10. At these high viscosities, the simulation time step approaches zero, significantly increasing computational time. Using the Papanastasiou (1987) approach, the regularised viscosity used in simulations is:

$$\hat{\eta} = \frac{\tau_y}{\gamma}\left(1 - e^{-c\dot{\gamma}}\right) + \mu + \alpha\dot{\gamma} \tag{11}$$

where $c$ is the viscosity scaling parameter, larger values of $c$ result in a better approximation of the constitutive equation (equation 10), while smaller values result in smaller apparent viscosities and larger simulation time steps. Here we set $m = 200$, a value which yielded the best balance between simulation speed and accuracy from validation simulations.

**Lahar simulations**

Static and dynamic pressures acting on the buildings in the Quebrada Dahlia study area were determined for twelve different inundation scenarios. The scenarios were designed to explore a wide range of flow types and





velocities and therefore may not represent any specific event or plausible set of events likely in Quebrada Dahlia. We use the same SPH particle spacing and terrain resolution (12.5 cm) of previous simulations by Mead et al.

(2015). This resolution provided the best balance between computational time and resolution of fine scale features that can affect the flows.

Simulations were run at constant flow rates of 25, 50, 75 and 100 $m^3s^{-1}$ for three different flow types in order to determine the effect of rheology and flow velocities on flow dynamics and forces exerted on buildings. The ratio between inertial and gravitational forces, expressed through the Froude number, was kept below 1 (subcritical

flow) for each flow rate by varying the inflow area. Froude number consistency was used here as inertial and gravitational forces are dominant controls on environmental flows such as these. Flow types were selected to represent the characteristics of the most commonly occurring flows in Arequipa – flash flood, hyperconcentrated streamflow and fine-grained, matrix-supported debris flow (Thouret et al., 2013). Rheology of flash flood flows was considered to be completely Newtonian with a viscosity of water (i.e. $\tau_y$, $\alpha = 0$, $\mu = 0.001$ and $\rho = 1000$),

rheological parameters for hyperconcentrated and debris flows (Table 2) were chosen using the dimensionless ratio between dispersive and viscous stresses explained in Julien and Lan (1991). Values of $\tau_y$, $\mu$ and $\alpha$ were taken from the experiments of Govier et al. (1957) and Bagnold (1954), reported in Julien and Lan (1991). For a hyperconcentrated streamflow, we presumed a particle concentration by volume ($C_v$) of approximately 30% consisting mostly of finer particles, meaning viscous stresses are still relatively important. Debris flow scenarios

were assumed to contain larger particles at a higher $C_v$ of approximately 55%. While the higher concentration increases the viscosity compared to a hyper-concentrated flow, the dispersive stress coefficient is also much higher, meaning that dispersive stresses will have more importance in determining flow behaviour.

The first 45 seconds of lahar flow were analysed for each scenario. While the flow was not established and constant by 45 seconds, computational cost limited simulation duration, so the scenarios considered here are more

representative of the damage caused by an initial lahar surge. We expect these surges cause the most damage to buildings as they have higher velocities and depths than a steady lahar flow.

**Flow behaviour**

Figure 5 displays snapshots of velocity and dynamic pressure magnitudes for each flow type at a flow rate of 75 $m^3s^{-1}$. Snapshots were taken at 15-second intervals and dynamic pressure was calculated as $\rho v^2$. Lahars mostly

followed the developed channel of Quebrada Dahlia for the first 15 seconds before overtopping the bank and spreading outwards. The difference between Newtonian and hyper-concentrated flow velocity profiles is minimal. While there is a small yield and dispersive stress component to the hyperconcentrated rheology, the Newtonian component (i.e. the linear stress-strain relationship) still appears to be important for these flows. However, the higher density of hyperconcentrated rheologies compared to Newtonian causes an observable difference in the

magnitude of dynamic pressure. Velocities are much lower for the debris flow rheology, presumably as a result of the higher dispersive coefficient. The maximum pressure is still similar to that of Newtonian and hyperconcentrated flows as maximum velocities are mostly confined to the channel.

The highest dynamic pressure magnitudes in Fig. 5 are shown along the centre of the channel, with much lower pressures in the vicinity of the buildings. Dynamic pressure may therefore not be acting perpendicular to the city

blocks. The directional components of dynamic pressure are important to consider as the critical strength of a wall or building is determined from forces acting normal (perpendicular) to the structure. Figure 6 shows the directional





components of dynamic pressure at 40 seconds for a flow rate of 75 m³s⁻¹. The section of Quebrada Dahlia studied here runs in a North-South direction and the buildings are oriented parallel to the channel, so a broad understanding of normal pressure acting on buildings can be interpreted from the NS and EW components of pressure. Figure 6

shows a consistent pattern across the rheology range where the pressure magnitude is dominated by the pressure acting in the streamwise (N-S) direction. The perpendicular pressure applied to walls facing the stream (~E-W pressure) is much lower than the streamwise pressure. Higher EW pressures are observed along cross streets splitting each city block; however, the pressure component acting perpendicular (NS) to these walls is also minimal. These observations indicate that the pressure magnitude, which is often assumed to be acting

perpendicular to walls (e.g. Zanchetta et al., 2004; Jenkins et al., 2015), can be much higher than normal pressure acting on walls and the use of pressure magnitude could therefore lead to an over-estimation of building damage. Figure 7 compares the mean pressure magnitude and the mean normal pressure acting on the 'West 2' block (see Fig. 2c) for various flow types. The pressures are measured for walls oriented approximately parallel to the quebrada (labelled 'Parallel') and north facing walls that are oriented approximately perpendicular to the quebrada

(labelled 'Perpendicular'). The normal pressures exerted on parallel walls are up to five times lower than the pressure magnitude. Normal pressure applied to perpendicular walls also differ from the pressure magnitude and the timing of peak mean pressure is affected. This further indicates the importance of considering normal pressure components when estimating damage.

Mean normal pressures acting on each block in the study area are shown in Fig. 8 for a flow rate of 75 m³s⁻¹.

Blocks 'East 1' and 'West 1' do not have walls facing perpendicular to the flow and therefore have no pressures recorded in that orientation. The pressure for each block generally follows a similar pattern through time with a well-defined peak pressure and a lower, steady background pressure. The rise of pressure to its peak value and reduction to its background value occurs over the space of approximately 20 seconds for each block and is likely the result of an initial surge of flow. This timeframe is too short to allow for an equalisation of hydrostatic pressure

between the inside and outside of buildings, suggesting that both hydrostatic and dynamic pressures are acting on walls during lahar surges. The timing of the peak is delayed for downstream blocks and the magnitude of the peak for each block varies. The differences in peak pressure are caused by exposure effects such as orientation and elevation of each block relative to the quebrada. Walls facing perpendicular to the stream are generally exposed to higher pressures than parallel walls, but this effect appears to vary and could be dependent on elevation

differences.

In terms of rheology, hyperconcentrated flows displayed the highest dynamic pressures acting on blocks as the higher density causes larger dynamic pressures. This effect is moderated by the yield strength of the hyperconcentrated flows, which cause the velocity (and therefore pressure) to be lower than Newtonian flows near perpendicular walls. Pressures for the debris flow are much lower than Newtonian and hyperconcentrated flows

due to the yield strength and significant dilatant component that limits velocities outside of the main channel.

### Application of critical depth-pressure curves

Depth at the maximum value of mean pressure along the block for each scenario is used to determine if the buildings in the study area can withstand the bending moment applied by hydrostatic and dynamic pressure. Figures 9 to 11 plot the peak pressure and 'surge depth' (depth at the time of peak pressure) for Newtonian,

hyperconcentrated and debris flows alongside critical depth-pressure curves for vulnerability classes A0, A and



B with a brick width of 150 mm (results for 250 mm brick widths are provided as supplementary material). The hazard variables of flow rate and lahar rheology appear to have an effect on building damage, although the size of the effect is difficult to determine since most scenarios place depth and pressure combinations well above the critical curves for each block. The flow depth, which affects hydrostatic pressure and bending moment location,

generally increases with the flow rate while the dynamic pressure appears to be mostly controlled by the rheology in combination with flow rate. The forces applied to the 'West 2' block, containing one class A and 4 class B buildings, are lower than the other blocks. This is possibly due to the blocks relative elevation and orientation to the quebrada (i.e. exposure) affecting dynamic pressure and fluid height. Debris flow scenarios at flow rates of 25, 50 and 75 $m^3s^{-1}$ indicate depths and pressures below the critical limit for this block's building classes.

The orientation of walls to the flow direction is another element of exposure that affects the normal pressure exerted on walls. In a number of scenarios, perpendicular walls are subjected to higher dynamic pressures and lower depths than parallel walls. However, this affect appears to be conditional to the rheology of the flow as the opposite is true for debris flow scenarios. These two effects demonstrate the importance of considering exposure elements separately to vulnerability as the hazard causes the effect of vulnerability to vary.

The proportion of buildings with depths and pressures above the critical curve for each scenario is shown in Fig. 12 for 0.15 m brick widths and Fig. 13 for 0.25 m brick widths. Assuming a binary damage state model where damage is complete for depths–pressure combinations above the curve, these proportions can be used to directly represent building loss. For the thinner bricks, all class A0 buildings are above the curve for all scenarios with the exception of the 25 $m^3s^{-1}$ debris flow. The 'East 1' block is not inundated in this scenario, resulting in two

undamaged class A0 buildings. Class A and B buildings are also mostly destroyed, with the exception of lower flow rate hyperconcentrated and debris flow scenarios where some blocks are on the edge of inundation and therefore subjected to much lower depth–pressure combinations. Slightly fewer building losses occur with larger brick widths (Fig. 13) as the larger section modulus results in a greater resistance to bending moments. However, most buildings are still destroyed in Newtonian and hyperconcentrated flow scenarios. An exception to this is the

75 $m^3s^{-1}$ Newtonian flow where the highest pressure on the 'East 1' block occurs early in the simulation when the surge depth is low, reducing the size of the applied moment.

The building loss results indicate that class A0 buildings are most vulnerable, with class A buildings marginally stronger due to the roof support. Losses for type B buildings in this area are much lower; however, this appears to be more related to building exposure than structural strength as most type B buildings are located in two blocks

subjected to lower depth–pressure combinations across scenarios. Overall, the data presented here suggests that building strength (i.e. the vulnerability component) has a minimal effect on losses and building location (i.e. exposure) relative to flow rate and type (i.e. hazard) plays a much greater role in determining loss.

**Discussion**

Losses in Figs. 12 and 13 assume damage is complete for depths and pressures above the critical curve. The

critical depth-pressure curve is the contour where the ratio of applied (pressure) moment equals the ultimate (failure) moment. Given that only direct actions are considered in this study, the curves likely form an upper bound to complete damage and depth–pressure combinations below the curve may still result in building damage through other mechanisms. While direct actions are regarded as the most important source of damage, they are also favoured in risk assessment due to the large scale predictability of hydrostatic and dynamic forces (Kelman





and Spence, 2004). Damage is likely to also be caused by scour and large debris missiles within the flow (Jenkins et al., 2015). In particular, boulders are often carried by lahars at the flow front (Iverson, 1997; Doyle et al., 2011) and can lead to significant damage (e.g. Zeng et al., 2015). However; these actions are harder to predict and incorporate into large scale loss analyses (Kelman and Spence, 2004). These unstudied actions are generally proportional to depth, pressure or velocity, indicating that there may be a relationship between the ratio of applied

to ultimate moment and damage through other actions. However, comprehensive data on loss events would be required to accurately refine the damage state model into several different damage states that consider the effect of other actions.

Pressure surges observed in the simulation occurred over too short a duration to allow for equalisation of hydrostatic pressure between the inside and outside of buildings. As a result, both hydrostatic and dynamic

pressures were considered in bending moment calculations. Slower increases in depth or buildings with a large number of openings could result in an equalisation of water depths and cause the effect of hydrostatic pressure to be negligible. However, lahar depth would still be an important factor to consider in building damage estimation as it still controls location of the bending moment and can cause damage through other actions (e.g. inundation damage, buoyancy, corrosion).

The applied depth at the time of maximum pressure was used here to create the depth-pressure combinations to determine building loss. This 'surge depth' was not necessarily the maximum depth of the lahar during the simulation. Maximum depths generally occurred at later times in the simulations when hydrostatic pressure may have equalised inside and outside buildings. This assumption was valid for most cases, although the losses for the 75 $m^3s^{-1}$ Newtonian flows indicate that this approach can be too simplistic at times. The complexity of lahar flow

within urban environments with intricate geometry and obstacles similar to the case study area means that broad generalisations and assumptions about flow dynamics, such as the assumption of a 'surge depth', are often limited in their validity. The building type, flow rate and flow type appear to have a large effect on overall building losses (Figs. 12 and 13); however, the variability in individual building losses appears to be predominantly caused by flow dynamics and building exposure (e.g. proportion of building types and orientation within blocks). This

suggests that urban flow environments may be too complicated to directly estimate flow behaviour from observed building losses.

**Conclusion**

Development of fragility functions in the form of critical depth-pressure curves for building classes within Arequipa have helped to provide insight into possible building losses and their cause. The almost total simulated

building loss for all scenarios indicates that that the quality of buildings is insufficient in this area and that substantial losses can be expected in the event of inundation. As inundation level is controlled by lahar volume, rheology and building exposure, these factors are therefore the most important in determining damage in this study area. Vulnerability can be decreased by increasing the structural strength of buildings through retrofitting building structures to improve quality. Specific improvements such as adding roof support and utilising reinforced frames

comprised of equally spaced RC columns will increase the overall strength of buildings by reducing the slenderness ratio (equation 6). Wider masonry units (brick width) and stronger mortar joints will also increase the overall building strength by increasing wall stiffness and therefore resistance to bending moments. However, the





increased structural strength appears to only reduce losses in very low flow rate scenarios where there is proportionally less inundation.

The approach demonstrated here, while focusing on building typologies in Arequipa, can be generalised to quantify masonry building loss in terms of flow depth and pressure in other areas. However, sufficient data on building strength is often not readily available on a large scale and demonstrates the need for focused studies in high risk areas affected by lahars. This also highlights the complementary relationship between large-scale vulnerability indices and direct vulnerability-damage relationships. Large-scale indicators of vulnerability can

locate areas in need of focused study and loss analysis. Detailed modelling of vulnerability, shown here, can be used in turn to refine the indices in order to focus on the most relevant indicators of damage.

**Acknowledgements**

The authors would like to thank the Civil Defence office in Arequipa (Instituto Nacional de Defensa Civil INDECI), in particular the Regional office (Mrs A. Arguedas) and the Provincial/City office (Mr. J. Vasquez) for

support during field work as well as the students of the department of geology of the University Nacional San Aguistin in Arequipa. JCT's work in Arequipa has been supported by the Labex CLERVOLC (contribution number XXX), the PICS CNRS program and the French Embassy in Lima.

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



**Figures**

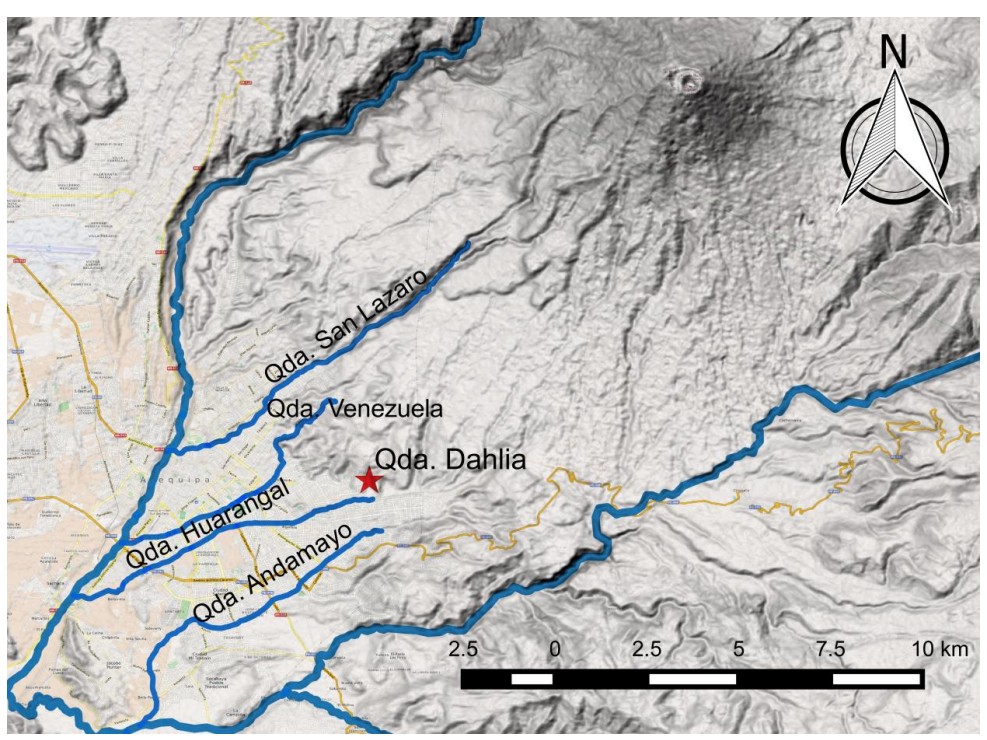


**Figure 1. Location of Arequipa in relation to El Misti volcano, showing the main quebradas and the location of the Quebrada Dahlia study area.**


Natural Hazards
and Earth System
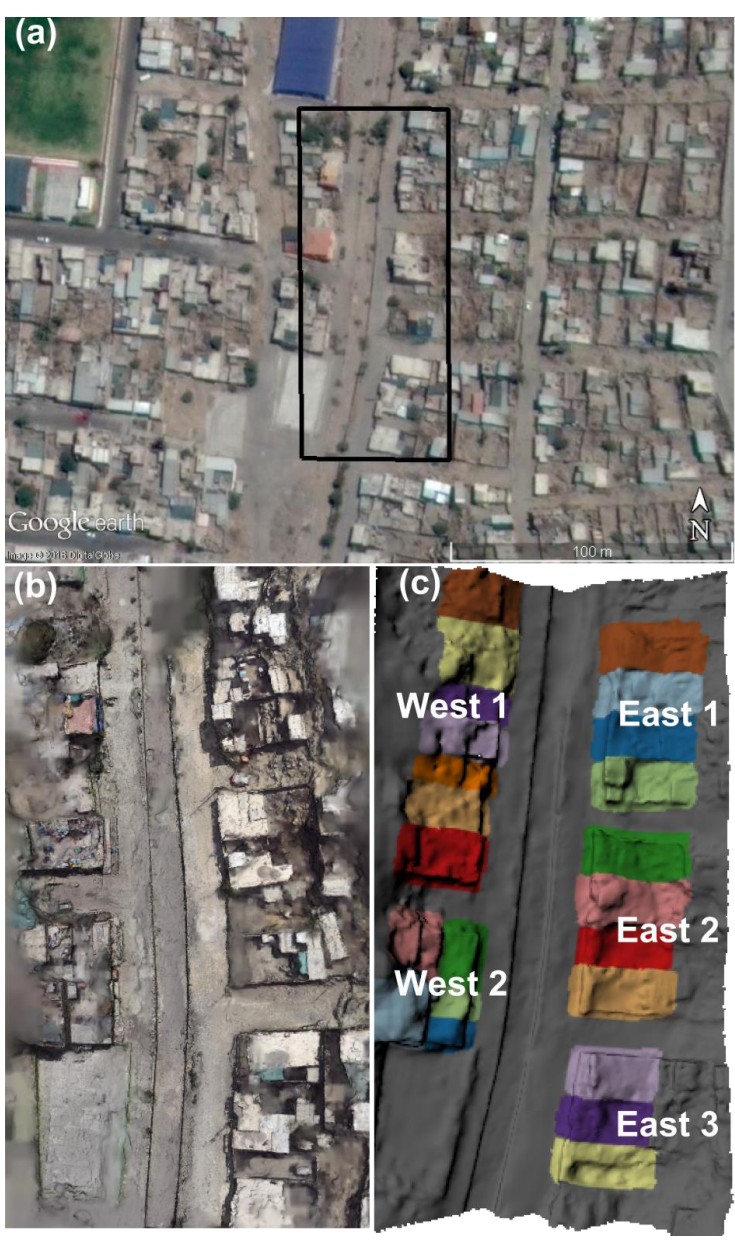

**Figure 2. Overview of Quebrada Dahlia study area, Arequipa, (a) aerial image with black outline showing study area, (b) photogrammetric reconstruction of the surface and (c) individual buildings and building blocks identified from building surveys.**






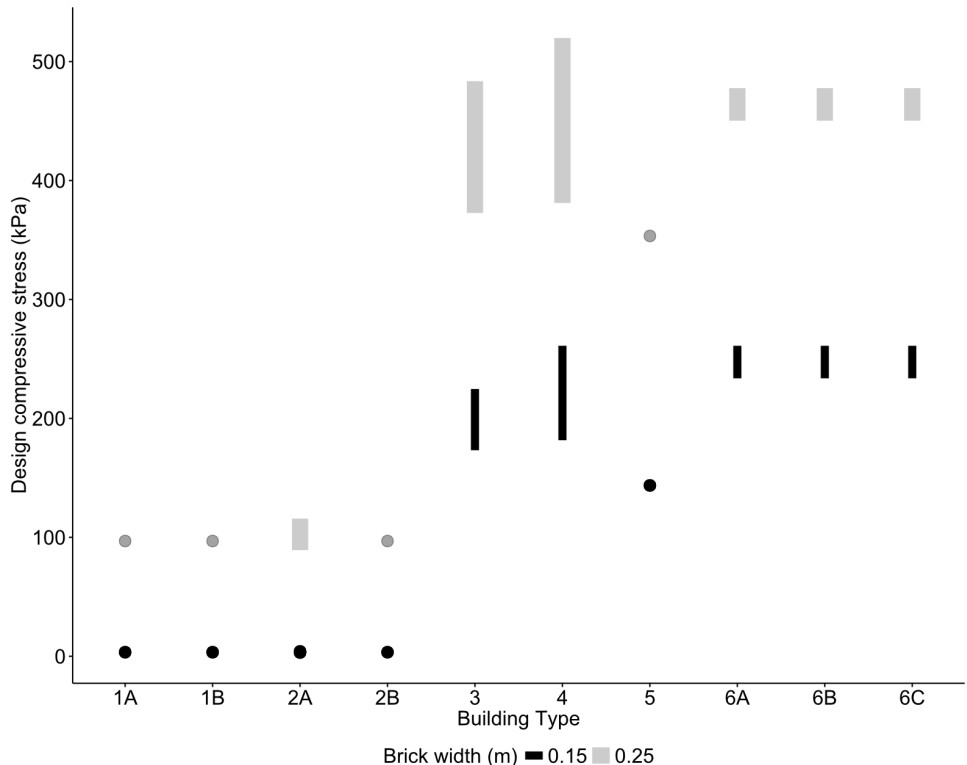

**Figure 3. Range of design compressive stress for building types 1A - 6C defined in Thouret et al. (2014). Compressive stress capacity was calculated for every configuration of compressive strength ($f_c$), bedded area ($A_b$), thickness coefficient ($k_t$) at brick widths ($b$) of 150 mm and 250 mm.**





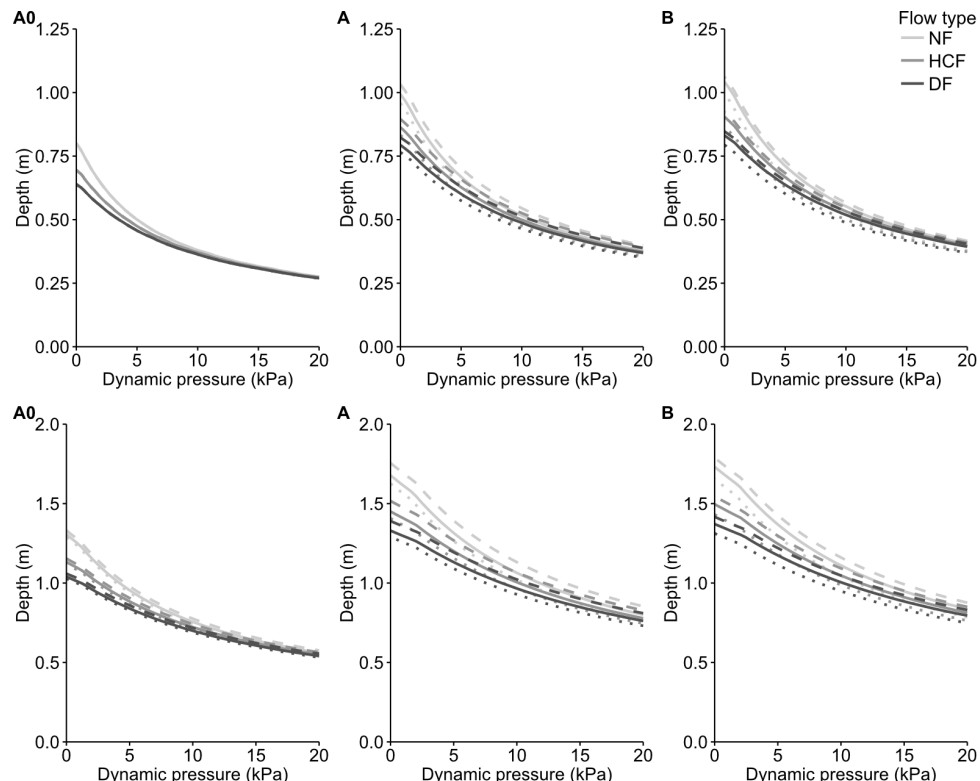

**Figure 4. Critical depth and dynamic pressures for failure of building classes A0, A and B for brick widths of 0.15 m (top) and 0.25 m (bottom). Shading of the lines indicate flow type and density, dotted lines and dashed lines represent the minimum and maximum forces required. Densities are for a Newtonian flow (NF, $\rho$ = 1000 kg·m$^{-3}$), hyper-concentrated flow (HCF, $\rho$ = 1500 kg·m$^{-3}$) and debris flow (DF, $\rho$ = 1915 kg·m$^{-3}$).**





**Figure 5. Evolution of dynamic pressure and velocity magnitudes for a 75 m³s⁻¹ flow along Quebrada Dahlia for a Newtonian flow (NF), hyperconcentrated flow (HCF) and debris flow (DF).**



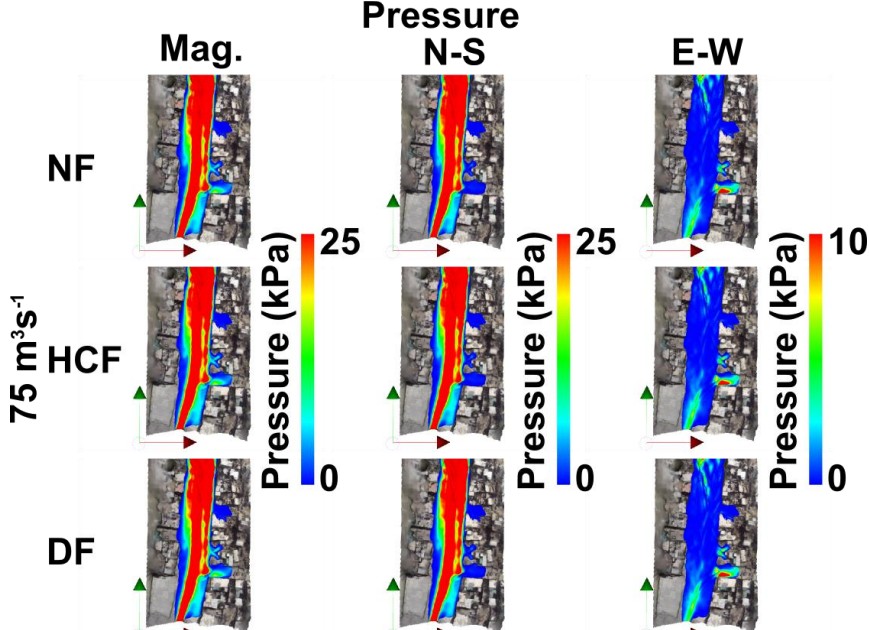

**Figure 6. Directional components of dynamic pressure for a 75 m$^3$s$^{-1}$ flow along Quebrada Dahlia for a Newtonian flow (NF), hyperconcentrated flow (HCF) and debris flow (DF).**



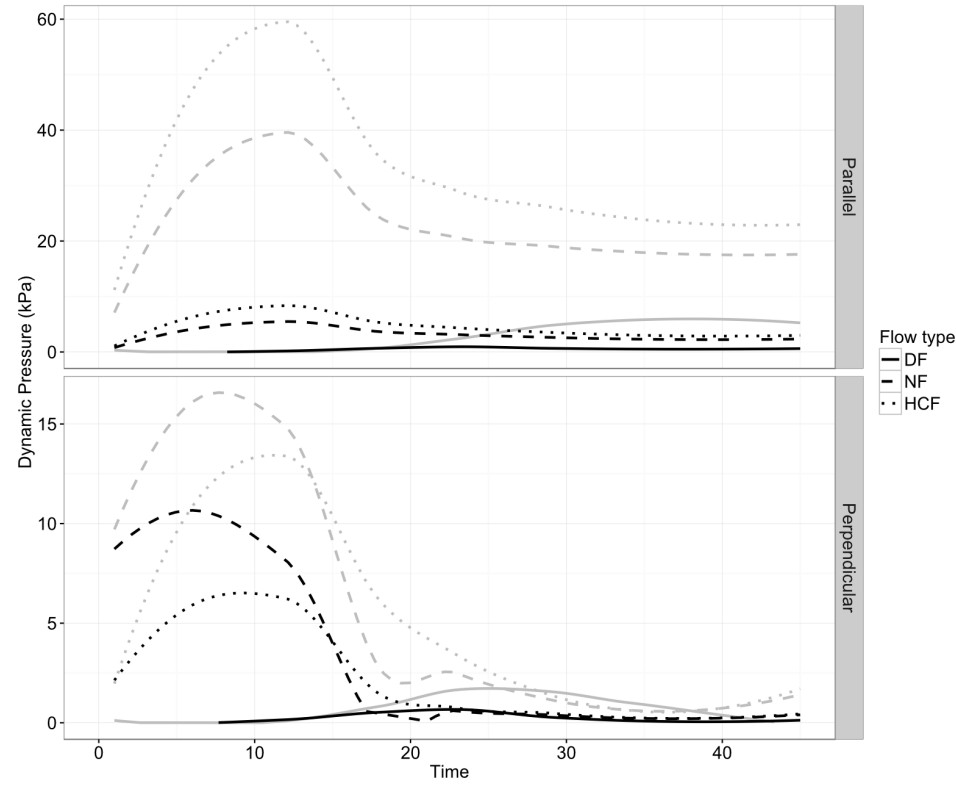

**Figure 7. Comparison of mean pressure magnitude (grey lines) and mean normal pressure (black lines) on block 'West 2' in the parallel and perpendicular orientations for a 75 m³s⁻¹ flow along Quebrada Dahlia.**




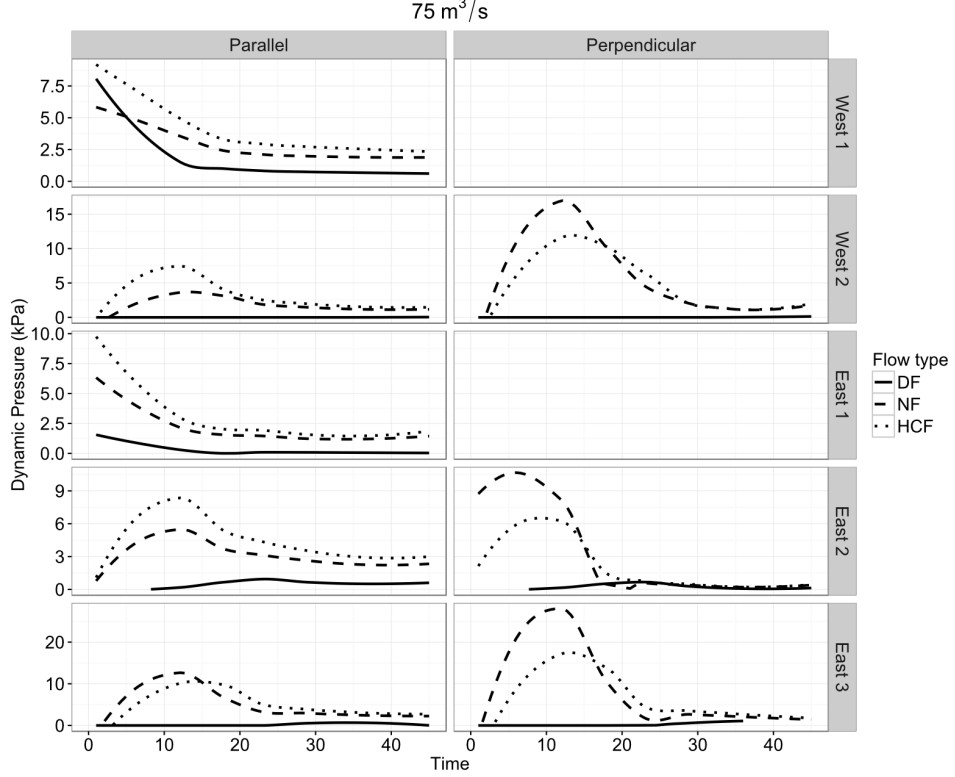

**Figure 8. Mean normal pressures applied to each city block in the perpendicular and parallel orientations for a 75 m³s⁻¹ flow.**






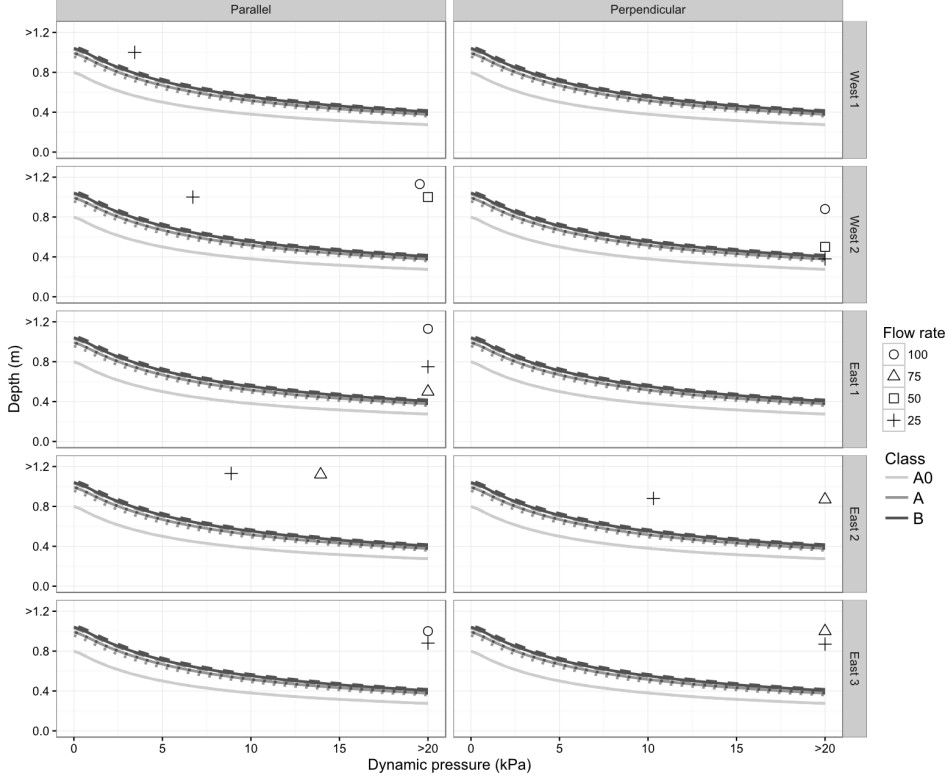

**Figure 9. Critical depth-pressure curves for building classes A0, A and B subjected to Newtonian flow. Peak normal pressures and corresponding depths applied to each city block are plotted as points for each flow rate.**





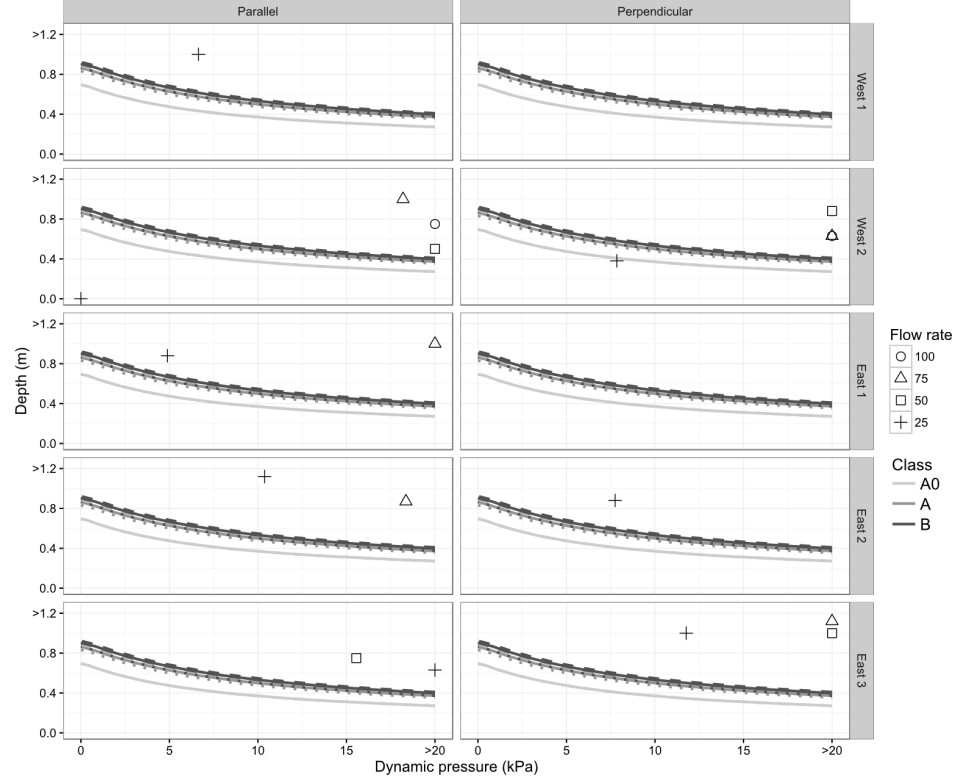


**Figure 10. Critical depth-pressure curves for building classes A0, A and B subjected to a hyperconcentrated flow. Peak normal pressures and corresponding depths applied to each city block are plotted as points for each flow rate.**





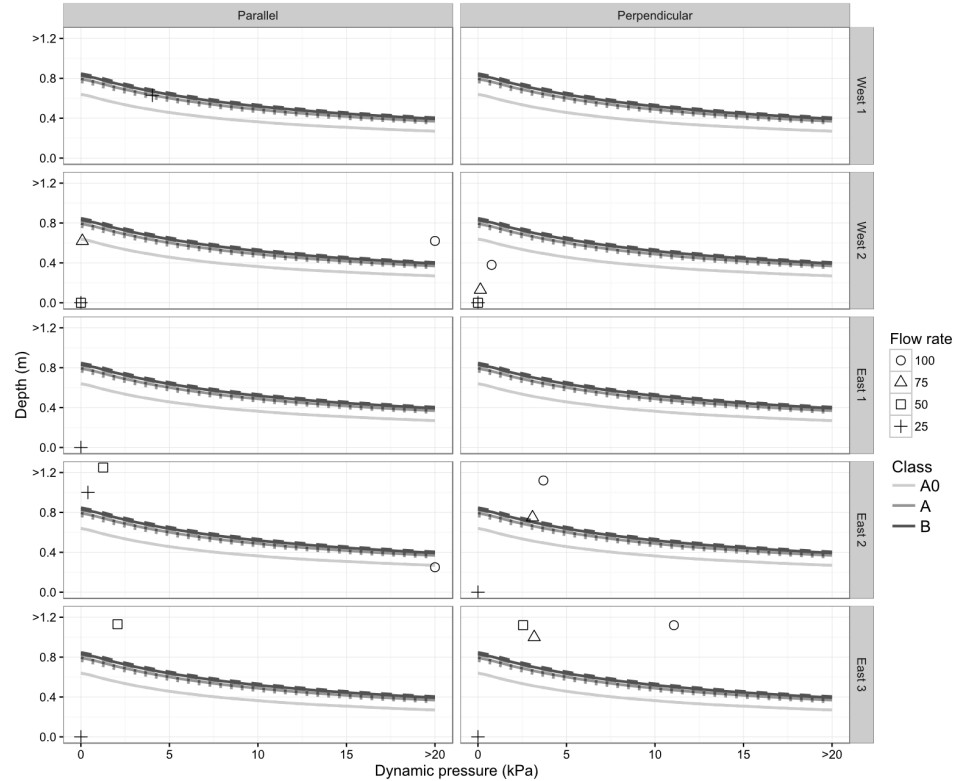

Figure 11. Critical depth-pressure curves for building classes A0, A and B subjected to a debris flow. Peak normal pressures and corresponding depths applied to each city block are plotted as points for each flow rate.



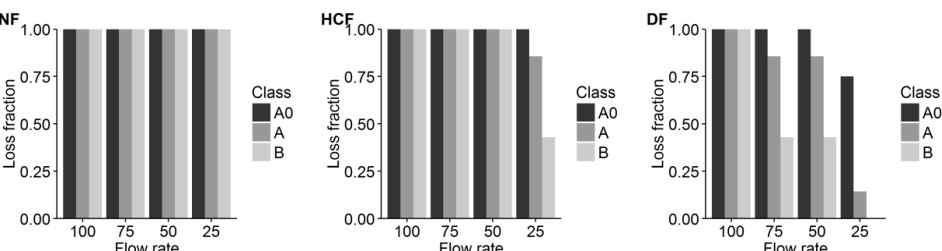

**Figure 12. Building loss fraction for all flow scenarios where buildings assumed to have a brick width of 0.15 m.**


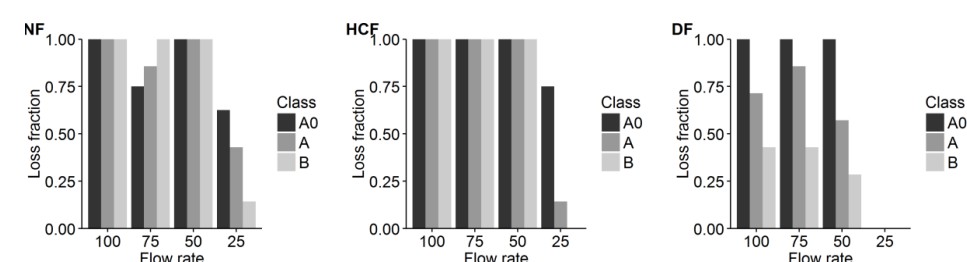

**Figure 13. Building loss fraction for all flow scenarios where buildings assumed to have a brick width of 0.25 m.**



**Tables**

**Table 1. Individual building type and vulnerability classes for each block in the Quebrada Dahlia study area. Block ID increases from north to south.**

| Block | ID | Type | Vulnerability Class | Vulnerability Class | Type | ID | Block |
|---|---|---|---|---|---|---|---|
| | 1 | 1A | A | A0 | 2A | 1 | |
| | 2 | 1A | A0 | A0 | 2A | 2 | East 1 |
| | 3 | 4 | B | A | 3 | 3 | |
| West 1 | 4 | 4 | A0 | A0 | 1B | 1 | |
| | 5 | 2B | A0 | A | 3 | 2 | East 2 |
| | 6 | 3 | A | A0 | 1B | 3 | |
| | 7 | 4 | B | A0 | 1A | 4 | |
| | 1 | 4 | B | B | 4 | 1 | |
| | 2 | 4 | B | A | 3 | 2 | |
| West 2 | 3 | 4 | B | A | 3 | 3 | East 3 |
| | 4 | 4 | B | B | 4 | 4 | |
| | 5 | 3 | A | | | | |

**Table 2. Density, particle concentration and rheology coefficients for hyperconcentrated streamflow and debris flow**
**simulations**

| Flow type | Density (kg m$^{-3}$) | Particle concentration by volume (%) | Yield strength ($\tau_y$, Pa) | Viscosity ($\mu$, Pa s) | Dispersive stress coefficient ($\alpha$) |
|---|---|---|---|---|---|
| Hyperconcentrated streamflow | 1500 | 30.3 | 0.94 | 0.0137 | $1.28 \times 10^{-5}$ |
| Fine-grained, matrix supported debris flow | 1915 | 55.5 | 0.672 | 0.0485 | 0.00224 |