# Peer review of "Examining the impact of lahars on buildings using numerical modelling"

_Natural Hazards and Earth System Sciences, 2016_

## Referee Comment (RC1) · Anonymous Referee #1 · 10 Oct 2016

General comments

Title: After having read the article, I do not think this study really quantifies the damage. Rather, it describes the conditions under which damage occurs, all the more that the authors consider either total damage or no damage, but no intermediate levels.

The building vulnerability part presents interesting results and could be interesting for the public of the journal but, in my opinion, some points could be explained better before considering the paper for publication. Moreover, I think that more warnings should be presented in the text each time where important assumptions are done and also the results, even though interesting, should be presented with more caution.

Overall, the article presents an interesting approach and tries to push forward research which in this topic is still very poor. However, this study relies on a patchwork of data

taken from other studies and parameters extrapolated from other contexts. The authors themselves state that lahar modeling scenarios may not represent any specific event or plausible set of events likely to happen in the studied area (line 268). Many assumptions guide the choice of decisive parameters and also the interpretation of results. As mentioned above, these are not always clearly pointed out nor thoroughly discussed.

The structure of the article consists of several parts that are in my opinion not enough interconnected. This makes it sometimes difficult to follow as a lot of different aspects are treated (civil engineering aspects to lahar rheology and modeling) with in each section quite an amount in technical vocabulary that not all readers will be familiar with. Important and recurrent terms such as vulnerability, hydrostatic, dynamic pressure, Newtonian, non-Newtonian, pressure magnitude, etc. should be clearly defined as they are used in different contexts and sections. The number of presented equations might be reduced and/or their presentation simplified as these are not key aspects of your study.

If this manuscript is accepted for publication, I am recommending major revision. The revision should aim at rendering the manuscript easier to read, better interweave the different sections to clearly show the relationships between the different considered aspects and parameters. Add a section where you clearly state all the assumptions you make and how these influence the credibility of your results. Add a section on limits and perspectives of this work. Ideally, the approach should be applied elsewhere and results compared to this study.

Specific comments

Building vulnerability: A general confusion is created by the use of building types from other studies (Thouret et al., 2013, 2014) which are referred to as building vulnerabilities. Clearly define what you mean by vulnerability. In my understanding, the classes you use from Thouret et al. are not indicating a vulnerability per se but correspond to different building types. If this is not the case, you need to explain how you integrate

the vulnerability calculation realized by Thouret et al. in your calculation. Is your result then a cumulated vulnerability? What is the weight of the different vulnerability indicators etc. Also, a minimum of explanations concerning these building types is necessary and it would be interesting to have a performance curve associated for each building type at the end.

Building vs block: it is not always clear throughout the paper what you are taking into account for your modeling and how data is aggregated. Are you using individual buildings? If yes, how many per block, what is the composition of each block in terms of building types, number, exposition, etc. If you use blocks, how is the "block" data generated from individual building data? Statistically? Subjective choice? There is more explanation needed here.

Lahar rheology and modeling: A lot of detailed description is provided on lahar rheology with the result that the reader remains relatively confused facing a complex topic with lots of technical terms in a few paragraphs. It is not clear from the structure of the text, what objective this section serves. A reorganization of this section might help to outline more specifically why the SPH model has been chosen, what its differences are compared to more commonly used lahar modeling software and what interest the use of this model has in terms of creating the depth-pressure curves.

Structural failure model similar to those employed by Roos (2003): Roos (2003) made his classifications and his study (comparison of the loads on the structure with the strength of the structures) for masonry buildings in the Netherlands, so the transposition for Arequipa buildings should be argued.

Technical corrections

Title Better : lahar induced damage

Abstract

Line 13 : uncertainty in number and type of buildings – this parameter is in my opinion

not to put in relation with potential building damage and its quantification. This is a question of scale and your objectives. What you may be rather referring to is the varying number and types of buildings, but uncertainty is misleading.

Line 15: What is the "relative importance" of lahar hazard, etc.?

Introduction

Line 24/25: may add here Vallance (2000) as a reference

Line 28: again, careful with the word "uncertainty" here, I think it is not adequate.

Line 34: clarify "relative" – is this from a statistical point of view?

Line 75: "to identify strategies that may reduce building loss" – these strategies appear to be mentioned only in a short paragraph in the conclusion which appears quite disconnected from the rest. If this really is one of your objectives, this needs more argumentation and be put in relation in a more concrete way with your modeling results.

Case study

Line 86: watch the spelling – torrenteRAS, one R at the end

Line 93: cite also and in first place Martelli 2011

Line 109: "relative effects" – again, don't understand the use of the word relative here. Be more precise.

Line 118: I don't understand the use of "by" here before cross streets. Does this mean the cross streets and quebrada separate the buildings in a way that blocks are formed?

Line 119: the "general approach" needs to be explained at least in a few sentences for those who do not have the time to read Thouret et al (2014).

Developing building vulnerability relationships

Line 133 and 135: Need to give more detail on these models/approaches.

Line 134-135: this is a very important assumption "Stresses for buildings in Arequipa are calculated using the approach specified in Australian Standard (AS) 3700-2011" and some better explanations should be provided

Line 135-136: Reformulate the sentence; it is not clear what you mean here.

Line 136: "In these models" – are you referring to the structural failure models?

Line137: "While some specifications in the standard may not be relevant for Arequipa, the calculation method is still valid for the area provided construction material properties from Arequipa are used as inputs." Yes, some specifications in the standard are not relevant for Arequipa, and the calculation method are not just related to the material properties, but also to the construction mode and the behavior, so "the methods are still valid" is not necessarily obvious, but should be argue in this way. Also important simplification or assumption "In these models, masonry walls are presumed to fail when the applied bending moment and shear forces are greater than the calculated ultimate bending moment and shear force the walls can withstand. We only consider the maximum bending moment here as preliminary investigations suggested the force required to overcome the ultimate moment was consistently lower than the force required overcoming the ultimate shear force".

This should be justified by literature investigation, experimental test or numerical simulation. The behavior of masonry wall is quite complex and can be (or generally is) a mix of flexural and shear behavior, in this case, considering just the maximum bending moment is an important assumption that can affect the global results and conclusions. See macroelement masonnery behaviour software TREMURI paper S. Lagomarsino, A. Penna, A. Galasco, S. Cattari, TREMURI program: an equivalent frame model for the nonlinear seismic analysis of masonry buildings, Eng Struct, 56 (2013), pp. 1787–1799 Moreover, when we use "design capacity specified in actual standards" for the constructions that have certainly "no design" or, in the better case "low code design", this affect the results and cannot be considered as representative for the studied area.

[Figure]

Line 139: "preliminary investigations" – of what? Lab tests? Modeling? Field work?

Line 146: "should not be greater than this value" – Why? Explain for those not familiar with this context.

Line 150 and following: quite a lot of equations and parameters presented. Since you are basically using parameter values specified in the Australian Standard, it may not be absolutely necessary to detail all of these equations . . .

Line 169 and following: a little graphic illustration would be SO helpful here!

Critical depth-pressure curves

Line 186 and following: I am not able to follow here, there are too many classes from different sources, I got lost. Can you present this in an easier way than A0, 1A-2B etc.? Or at least provide an informative table describing what these mean? General remark: you are analyzing vulnerability of buildings so it is a little strange to use already defined vulnerability classes. It would sound more logical to refer to simple building types or categories for the Thouret et al classifications instead of calling these vulnerability classes.

For figure 4, you mean that the graphic represents the combination between the depth and the dynamic pressures for which you have failure of the building class? This could be assimilated as a limiting right under the line you do not have collapse and above the line you have collapse? In order to verify the results for one structure or one wall it will be interesting to perform a kind of push-over curve (curve displacement-shear base response).

Maybe a table with the description of the building types from 1A to 6C would be useful for self-understanding of the paper accompanied by the building damage threshold for each typology (these thresholds are a very important issue, that can modify completely the results, and in my opinion this issue is not sufficiently treated in the paper).

Line 195 and following: OK, but is this representative? Most hyperconcentrated flows

in Arequipa carry boulders given the environment there, and the impact of those is very important although not taken into account at all in your study . . .

Lahar rheology

General remark: This is a long overview but it is not clear why you expose all this to introduce finally the quadratic rheology model. Rewrite this section illustrating from the beginning the quadratic rheology model and how it compares to other existent models and explain your choice rather than the general information which is rather disconnected from the rest. Also, you are using a lot of technical vocabulary here concerning the rheology which some part of the readers may not be familiar with. So, simplify and rather concentrate on providing definitions of critical terms such as non-Newtonian, Newtonian, etc.

Implementation in smoothed particle hydrodynamics

Line 246: OK, but you need to describe at least a little bit what this model is and how it is different from classically used lahar modeling software such as Titan2D/3D, LaharZ, etc.

Line 257f: Again, I got lost here – what is your point?

Line 263: Where is m in the equation? How many validation simulations did you use and what data was chosen for it?

Line 268: "may not represent any specific event . . ." So what is the use of it? This is not at its place here. You may have a general discussion section where such issues should be discussed, but at this point, the reader may not think it is useful to continue the reading.

Line 269: 12.5 cm resolution? At this resolution, the identified building blocks and houses should be much more accurate than what appears in Figure 2 and Figure 5!

Line 281: Again, remind what these stand for and why you can take the experimental

values for Arequipa.

Line 286: Hyperconcentrated flows can have high viscosities, no?

Line 289: 45 seconds: why? Computational time and resulting cost? Maybe optimize this paragraph.

Line 290f: I do not agree here. Lahar surges do not always have higher depths than a steady lahar flow. First, a steady lahar flow is not necessarily the opposite of a lahar surge but a different flow type/phase. Second, the surge can be quite small and there can be several small surges before the major lahar flow front arrives. It depends a lot on the location and the environmental conditions triggering the lahar. So careful here to not make it sound as a general rule.

Flow behavior

Line 297ff: Not clear. What is your point here?

Line 300: This is not new . . .

Line 300ff: Decreasing velocity can also be due to friction effects and turbulence along the channel walls

Line 308: "buildings oriented parallel to the channel": this depends how you define the long axis of the building. Looking at Figure 2, this is not very clear to me. What do you mean by "broad understanding of normal pressure"?

Line 313: illustrate this with a graphic illustration. "Higher EW pressures . . ." Not sure I understand what you mean. Flow looses velocity and depth when spreading in side-roads?

Line 316: Pressure magnitude: you need to define this term.

Line 323: normal pressure: need to define this term!

Line329: you were mentioning earlier that you consider only initial lahar surges, so this

is not really a conclusion of the modeling, is it?

Line 330: Why is this a consequence of the short timeframe?

Line 334: "higher pressures": you mean higher dynamic pressure?

Line 335: If this indicates elevation differences, it would be helpful to have a map illustrating what elevation differences there are in this study area.

Line 336: acting on blocks: rather buildings?

Line 337ff: reformulate, this paragraph is not clear. Also, I am skeptical that higher density generally causes larger dynamic pressure. . .can you provide reference?

Line 338: near perpendicular walls: and not near, what happens?

Line 340: The dilatant component is lower for debris flows than for Newtonian and hyperconcentrated flows, so the latter decrease in velocity quite rapidly when diminishing in depth which also reduces the pressure for these flows. . .

Application of critical depth-pressure curves

Line 342: "along the block" – means what?

Line 348f: Not clear, "well above the critical curves for each block". General remark: you switch between buildings and blocks and it is in the end not very clear for what you realize your modeling and how do you generate the curves for one block? Is it the mean value of all buildings contained in the block? In this case, we need to have more information on each block. . .

Line 353: fluid height – be consistent in the use of your terms to make it easier for the reader to follow. Either use fluid height or flow depth.

Line 354: Which means no damage to expect?

Line 357: lower depths? This is interesting: I would have thought that depth rises when flow impacts a perpendicular wall. At least this is what you see frequently with

floodmarks on house walls after flash floods or torrential floods in the Alps . . . it would be interesting to compare your methodology with another location and see if you find similar relationships.

Line 359: "effect of vulnerability": this means what?

Line 371: explain in more detail what this means "reducing the size of the applied moment".

Line 375: across scenarios: means that this can be observed in several scenarios? How many did you have in total? Other studies also indicate that distance from the channel plays a more important role than the building type itself. . .

Discussion

Line 380ff: Reformulate, this paragraph is not clear.

Line 389: "applied to ultimate moment and damage through other actions" – how are these defined?

Line 390: comprehensive data: like what?

Line 393: Repetition, see line 329

Line 395ff: depends on localization of the building respective to the channel: first row, second row, screening effect, etc.

Line 409: this is not new. . .

Line 410: "may be too complicated" – no, not necessarily, it just needs better assessments, more data and detailed field studies. . .

Conclusion

Line 416: not new . . .

Line 417ff: This paragraph comes somehow out of the blue and is quite disconnected

from the previous rather technical descriptions. The recommendations for retrofitting are not enough put in relation with the results of your modeling, it would maybe be a better option to add this into the discussion and link it more specifically to your results.

Line 425: here you say the approach can be generalized but given all the restrictions and constraints you have for Arequipa, this sounds rather unlikely. The best way would be to propose a section where you apply this approach to a different study area and shortly explain how you validate your approach. Or you need to explain more in detail why and how this approach can be generalized.

Line 429: Large-scale indicators: such as?

Line 430: In my opinion, you are not modeling vulnerability. You use building type data, do lahar modeling and cross this information to generate building performance curves . . .

Line 431: "refine the indices in order to focus": only if data is available as for now you use a lot of data extrapolated from other sources.

Figures

Figure 1: need to complete the legend indicating what the yellow and orange lines stand for as well as the orange areas (built area: extent of the city at present?); reduce the size of the North arrow. The NE end of the quebradas Venezuela, Huarangal and Andamayo are quite low and appear disconnected from the foothill of the volcano – is there a reason for this presentation?

Figure 2: I think you need to better define what you understand by building block. Since you refer a lot to the works of Thouret et al. 2013 and 2014, his definition of building blocks seems to be different from what you delineate here as a block. Clarify. Also, the delineation of your blocks is not easy to understand: for example in fig. 2B, bottom left, you distinguish three blocks dark green, light green and dark blue; on the opposite side of the road, the light violet block appears to be a single block although you might

also have distinguished at least two; same for the light green block further North on the right side . . . if so is your choice, at least explain how you determine.

Figure 3: title of legend "Design compressive stress" – means what? Is it the compressive stress? "Building Type" – no capital letter for type.

Figure 4: Maybe simplify where possible this group of graphics: no need to put the "depth" legend title everywhere; reduce the appearing scale numbers (no need to detail every 0.25, just indicate 0,5; 1 and leave the little markers for subunits). There is a lot of information presented, so simplify a maximum.

Figure 5: what do the arrows indicate on the lower left of each image? Images are quite small, text is a little too big. Might be helpful to have an image without any modeling result where you locate the channel limits, outline the buildings and indicate flow direction.

Figure 6: same remark as above; legend too big, no need to reproduce the same legend for all three image groups. The pressure value to the left is too isolated, difficult to understand what this means.

Figure 7: OK, very nice.

Figure 8: OK, very nice.

Figure 9, 10 and 11: Difficult to distinguish the individual curves: maybe fine lines of different colors would be easier to read here.

Figure 12 and 13: Legend title "loss fraction" is a little strange not associated with percentages . . . Flow rate needs a unit! For the rest: same remark as above: simplify.

Tables

Table 1: Associate table with a figure where these numbers can be seen.

Table 2: OK – I am not an expert in streamflow/lahar rheology, so I am not able to

evaluate the correctness of these parameters. However, I understand these parameters are not your own modeling results but used to model, so provide reference for the source of these values in this table.

---

## Referee Comment (RC2) · Anonymous Referee #2 · 9 Jan 2017

**Description**

In this paper, the authors try to estimate the impact of lahars on buildings, with application to the city of Arequipa, Peru. The study is carried out by using numerical simulations of lahars under different hypotheses (Newtonian regime, hyperconcentrated flow, or debris flow). The simulations are based on the SPH technique. The authors indicate that the main hazard is related to the bending of the building walls due to the normal force exerted by the lahar.

Main comment

The authors identify that both hydrostatic and hydrodynamic forces contribute to the bending moment (and eventually the collapse) of the walls of a building. The former is related to the flow depth whereas the latter is related to its momentum. For describing the momentum transported by the lahar, the authors introduce the concept of "dynamic pressure", defined by the authors as $\rho v^2$ (see line 294), and describe the components of the force as "directional components of dynamic pressure" (see eg: line 305). Although, I agree with the adopted phylosophy, I suggest to slightly modify the used nomenclature. In fact, the "dynamic pressure", is usually defined as $1/2\rho v^2$ and it is a scalar. It is used in the context of the Bernoulli theorem for describing the variations of the fluid pressure along a streamline. I propose that, the author adopt the concept of flux of momentum, which is a tensor defined as (see eg: Landau and Lifshitz, 1959):

$$\Pi = P\mathbf{n} + \rho\mathbf{v}(\mathbf{v} \cdot \mathbf{n}) \tag{1}$$

where $\rho$ is the density of the fluid, $\mathbf{n}$ is the unitary vector normal to a surface, and $\mathbf{v}$ is the flow velocity. Considering a surface normal to the flow (parallel to $\mathbf{n}$), the component of the flux of momentum perpendicular the surface becomes $P + \rho v^2$, whereas when the velocity of the fluid is parallel to the surface, the only component of the flux of momentum perpendicular to the surface is $P$. The flux of momentum is related to the force exerted by a wall to deviate or stop the flow.

Minor corrections

- In equation (1), $b$ is referred as the thickness of the bricks used in the wall (see line 145). Later, at line 147, $b$ is referred as the wall thickness. This is different for walls with a thickness greater that the width of a single brick. Do you consider only walls made of a single layer of bricks?

- Line 149: In this context, I suggest to specify that the "normal forces" are vertical forces.

- Line 170: Symbol $a \mapsto a_v$

- Line 190: "The critical height". Do you mean "The critical depth"?

- Line 223: Usually the coefficient $\mu$ (eq.7) is defined "viscosity" only when the exponent $m$ is equal to 1.

- Line 252: "equation 3". Perhaps you mean eq. (7) or (8)?

- Lines 256 and 262: "equation 4"? Perhaps eq.(7)? Please check.

- Line 305: The "directional components of the dynamic pressure ...". Pressure, and dynamic pressure are scalars (see above).

- Lines 315-323: Please check if the concepts of "normal stress" is more appropriate than "normal pressure component".

- Line 437: contribution number "XXX", please provide the number if necessary.

**References**

Landau, L. D. and Lifshitz, E. M.: Fluid Mechanics, vol. 6, Pergamon Press, 1st ed. edn., 1959.

---

## Author Comment (AC1) · 24 Feb 2017

We would like to thank the reviewer for their time spent in providing a detailed and constructive review of our manuscript, the comments will result in a considerably improved manuscript. We appreciate (and agree) with comments that the results and approach presented would be interesting to the public of the journal and concede that some assumptions were not clearly identified and discussed in our attempt to improve on this field of research. The authors believe this work is important in its demonstration of an approach to estimate damage (or, as pointed out 'conditions under which damage occurs') where data may be lacking. Out of necessity, this requires use of 'a patchwork of data' and extrapolation from other contexts.

We have developed a plan to address your broad comments on the number and con-

nectivity of aspects (engineering, rheology and modelling), defining key terms, possible simplifications and the need for a critical and comprehensive discussion on the limitations and assumptions of this work. Revision of the manuscript is ongoing; a list of major changes needed to address the comments as well as a response to all specific comments are provided in the attached document.

Broadly, we have reduced the amount of technical detail by moving a large section of the building vulnerability work to an appendix and greatly reduced the size of lahar modelling sections by removing unnecessary details. We have added a 'limitations' subsection to the discussion section which critically evaluates and justifies assumptions used in this work. The purpose of the manuscript has been clarified ("...investigate the effect of hazard (flow rate and rheology), exposure (building orientation) and vulnerability (building quality/type) components on building loss in Arequipa."), readability has been improved through ensuring consistency and clarity of key terms and we will expand the introduction to explain all aspects of the work and how they are connected to the main aim.

Please also note the supplement to this comment:
http://www.nat-hazards-earth-syst-sci-discuss.net/nhess-2016-282/nhess-2016-282-AC1-supplement.pdf

**Supplement:**

**Quantifying lahar damage using numerical modelling: Reviewer #1 response**

Stuart R. Mead[1,2], Christina Magill[1], Vincent Lemiale[2], Jean-Claude Thouret[3] and Mahesh Prakash[2]

[1]Risk Frontiers, Department of Environmental Science, Macquarie University, Sydney, Australia
[2]Commonwealth Scientific and Industrial Research Organisation, Clayton 3168, Victoria, Australia
[3]Laboratoire Magmas et Volcans UMR6524 CNRS, IRD and OPGC, University Blaise Pascal, Campus Les Cézeaux, 63178 Aubière , France

*Correspondence to:* Stuart R. Mead (Stuart.Mead@mq.edu.au)

**Response to Reviewer #1:**

*General comments*

Title: After having read the article, I do not think this study really quantifies the damage. Rather, it describes the conditions under which damage occurs, all the more that the authors consider either total damage or no damage, but no intermediate levels.

The building vulnerability part presents interesting results and could be interesting for the public of the journal but, in my opinion, some points could be explained better before considering the paper for publication. Moreover, I think that more warnings should be presented in the text each time where important assumptions are done and also the results, even though interesting, should be presented with more caution.

Overall, the article presents an interesting approach and tries to push forward research which in this topic is still very poor. However, this study relies on a patchwork of data taken from other studies and parameters extrapolated from other contexts. The authors themselves state that lahar modeling scenarios may not represent any specific event or plausible set of events likely to happen in the studied area (line 268). Many assumptions guide the choice of decisive parameters and also the interpretation of results. As mentioned above, these are not always clearly pointed out nor thoroughly discussed.

The structure of the article consists of several parts that are in my opinion not enough interconnected. This makes it sometimes difficult to follow as a lot of different aspects are treated (civil engineering aspects to lahar rheology and modeling) with in each section quite an amount in technical vocabulary that not all readers will be familiar with. Important and recurrent terms such as vulnerability, hydrostatic, dynamic pressure, Newtonian, non-Newtonian, pressure magnitude, etc. should be clearly defined as they are used in different contexts and sections.

The number of presented equations might be reduced and/or their presentation simplified as these are not key aspects of your study.

If this manuscript is accepted for publication, I am recommending major revision. The revision should aim at rendering the manuscript easier to read, better interweave the different sections to clearly show the relationships between the different considered aspects and parameters. Add a section where you clearly state all the assumptions you make and how these influence the credibility of your results. Add a section on limits and perspectives of this work. Ideally, the approach should be applied elsewhere and results compared to this study.

We would like to thank the reviewer for their time spent in providing a detailed and constructive review of our manuscript, the comments will result in a considerably improved manuscript. We appreciate (and agree) with comments that the results and approach presented would be interesting to the public of the journal and concede that some assumptions were not clearly identified and discussed in our attempt to improve on this field of research. The authors believe this work is important in its demonstration of an approach to estimate damage (or, as pointed out 'conditions under which damage occurs') where data may be lacking. Out of necessity, this requires use of 'a patchwork of data' and extrapolation from other contexts.

We have developed a plan to address your broad comments on the number and connectivity of aspects (engineering, rheology and modelling), defining key terms, possible simplifications and the need for a critical and comprehensive discussion on the limitations and assumptions of this work. Revision of the manuscript is ongoing; a list of major changes needed to address the comments as well as a response to all specific comments are provided in the attached document.

Broadly, we have reduced the amount of technical detail by moving a large section of the building vulnerability work to an appendix and greatly reduced the size of lahar modelling sections by removing unnecessary details. We have added a 'limitations' subsection to the discussion section which critically evaluates and justifies assumptions used in this work. The purpose of the manuscript has been clarified ("…investigate the effect of hazard (flow rate and rheology), exposure (building orientation) and vulnerability (building quality/type) components on building loss in Arequipa."), readability has been improved through ensuring consistency and clarity of key terms and we will expand the introduction to explain all aspects of the work and how they are connected to the main aim.

**Specific comments**

Building vulnerability: A general confusion is created by the use of building types from other studies (Thouret et Al., 2013, 2014) which are referred to as building vulnerabilities. Clearly define what you mean by vulnerability. In my understanding, the classes you use from Thouret et al. are not indicating a vulnerability per se but correspond to different building types. If this is not the case, you need to explain how you integrate the vulnerability calculation realized by Thouret et al. in your calculation. Is your result then a cumulated vulnerability? What is the weight of the different vulnerability indicators etc. Also, a minimum of explanations concerning these building types is necessary and it would be interesting to have a performance curve associated for each building type at the end.

Your understanding is correct here, the classes from Thouret et al. correspond to different building types. We are now referring to these as simplified structural classes. An additional table that provides a description of each building type and typology and their simplified structural class has been included.

Building vs block: it is not always clear throughout the paper what you are taking into account for your modeling and how data is aggregated. Are you using individual buildings? If yes, how many per block, what is the composition of each block in terms of building types, number, exposition, etc. If you use blocks, how is the "block" data generated from individual building data? Statistically? Subjective choice? There is more explanation needed here.

Building, block and block orientation will be comprehensively discussed in the Case study section, highlighting the relationship between them. Figure 2 (which will be modified) and Table 1 also help to explain the number of buildings per block, and composition.

80 Lahar rheology and modeling: A lot of detailed description is provided on lahar rheology with the result that the reader remains relatively confused facing a complex topic with lots of technical terms in a few paragraphs. It is not clear from the structure of the text, what objective this section serves. A reorganization of this section might help to outline more specifically why the SPH model has been chosen, what its differences are compared to more commonly used lahar modeling software and what interest the use of this model has in terms of creating the depth-

85 pressure curves.

We have reduced the amount of discussion on lahar rheology to focus only on the core details needed to understand this study. Additional explanation on SPH and why it was chosen over other approaches (also explained) will be provided.

90 Structural failure model similar to those employed by Roos (2003): Roos (2003) made his classifications and his study (comparison of the loads on the structure with the strength of the structures) for masonry buildings in the Netherlands, so the transposition for Arequipa buildings should be argued.

This is to be addressed in an expanded 'limitations and discussion' section.

95 **Technical corrections**

**Title**

Title Better: lahar induced damage

Title changed to 'Examining the impact of lahar on buildings using numerical modelling'

100 **Abstract**

Line 13: uncertainty in number and type of buildings – this parameter is in my opinion not to put in relation with potential building damage and its quantification. This is a question of scale and your objectives. What you may be rather referring to is the varying number and types of buildings, but uncertainty is misleading.

Changed to '…, varying number and type of buildings…".

105

Line 15: What is the "relative importance" of lahar hazard, etc.?

'Relative' as a word is irrelevant here, changed to "…examine the importance of lahar hazard..."

**Introduction**

110 Line 24/25: may add here Vallance (2000) as a reference

The reference to Vallance and Iverson (2015) is from the second edition of Encyclopaedia of Volcanoes were Vallance (2000) is from the first. They are functionally the same work.

Line 28: again, careful with the word "uncertainty" here, I think it is not adequate

115 Changed to '…varying number of elements…"

Line 34: clarify "relative" – is this from a statistical point of view?

See previous response, 'relative' is removed.

Line 75: "to identify strategies that may reduce building loss" – these strategies appear to be mentioned only in a short paragraph in the conclusion which appears quite disconnected from the rest. If this really is one of your objectives, this needs more argumentation and be put in relation in a more concrete way with your modeling results.

Rather than strategies to reduce loss, the aim was to investigate the effect of hazard (flow rate and rheology), exposure (building orientation) and vulnerability (building quality/type) components on building loss in Arequipa. This has been corrected and is consistent throughout the manuscript.

**Case Study**

Line 86: watch the spelling – torrenteRAS, one R at the end

Fixed throughout manuscript.

Line 93: cite also and in first place Martelli 2011

Added according to NHESS reference style.

Line 109: "relative effects" – again, don't understand the use of the word relative here. Be more precise.

Again, 'relative' is irrelevant in this context.

Line 118: I don't understand the use of "by" here before cross streets. Does this mean the cross streets and quebrada separate the buildings in a way that blocks are formed?

Modified figure 2a and reworded sentence to clearly explain how the buildings are separated into to five city blocks.

Line 119: the "general approach" needs to be explained at least in a few sentences for those who do not have the time to read Thouret et al (2014).

The following sentences explained the building classification method of Thouret et al. (2014), this has been reworded to make it clear.

**Developing building vulnerability relationships**

Line 133 and 135: Need to give more detail on these models/approaches.

A short summary of the methods on both approaches has been added.

Line 134-135: this is a very important assumption "Stresses for buildings in Arequipa are calculated using the approach specified in Australian Standard (AS) 3700-2011" and some better explanations should be provided.
Line 135-136: Reformulate the sentence; it is not clear what you mean here.
Line 136: "In these models" – are you referring to the structural failure models?

Line 137: "While some specifications in the standard may not be relevant for Arequipa, the calculation method is still valid for the area provided construction material properties from Arequipa are used as inputs." Yes, some specifications in the standard are not relevant for Arequipa, and the calculation method are not just related to the material properties, but also to the construction mode and the behavior, so "the methods are still valid" is not necessarily obvious, but should be argue in this way.

160

These comments have been dealt with simultaneously. A better explanation of why the standard can be relevant for Arequipa is provided, and the effects of this assumption is discussed in the reworked discussion section.

Also important simplification or assumption "In these models, masonry walls are presumed to fail when the applied bending moment and shear forces are greater than the calculated ultimate bending moment and shear force the walls can withstand. We only consider the maximum bending moment here as preliminary investigations suggested the force required to overcome the ultimate moment was consistently lower than the force required overcoming the ultimate shear force". This should be justified by literature investigation, experimental test or numerical simulation. The behavior of masonry wall is quite complex and can be (or generally is) a mix of flexural and shear behavior, in this case, considering just the maximum bending moment is an important assumption that can affect the global results and conclusions. See macroelement masonnery behaviour software TREMURI paper S. Lagomarsino, A. Penna, A. Galasco, S. Cattari, TREMURI program: an equivalent frame model for the nonlinear seismic analysis of masonry buildings, Eng Struct, 56 (2013), pp. 1787–1799.

165

170

Line 139: "preliminary investigations" – of what? Lab tests? Modeling? Field work?

175 The response to shear is important to consider, however the effect of this is minimal in our study. In an effort to simplify the manuscript we chose to leave discussion of shear out of the manuscript. However, for the reasons you identified, this was probably not the best idea. Instead, we have added additional information on the shear response in an appendix and made reference to it in this section. This way, the focus of the manuscript is maintained, but additional information is available to readers if needed.

180

Moreover, when we use "design capacity specified in actual standards" for the constructions that have certainly "no design" or, in the better case "low code design", this affect the results and cannot be considered as representative for the studied area.

This is addressed in the limitations and discussion section.

185

Line 146: "should not be greater than this value" – Why? Explain for those not familiar with this context.
Line 150 and following: quite a lot of equations and parameters presented. Since you are basically using parameter values specified in the Australian Standard, it may not

190 be absolutely necessary to detail all of these equations.
Line 169 and following: a little graphic illustration would be SO helpful here!

Details on calculation of ultimate moment (and now shear) are added to the appendix as these components have been described previously (e.g. Zheng et al. (2009)) and are in standards.

195 **Critical depth-pressure curves**

Line 186 and following: I am not able to follow here, there are too many classes from different sources, I got lost. Can you present this in an easier way than A0, 1A-2B etc.? Or at least provide an informative table describing what these mean?

A new table has been constructed with general descriptions to help understanding of this and the previous section.

200

General remark: you are analyzing vulnerability of buildings so it is a little strange to use already defined vulnerability classes. It would sound more logical to refer to simple building types or categories for the Thouret et al classifications instead of calling these vulnerability classes.

For clarity, we are calling these simplified structural classes throughout.

205

For figure 4, you mean that the graphic represents the combination between the depth and the dynamic pressures for which you have failure of the building class? This could be assimilated as a limiting right under the line you do not have collapse and above the line you have collapse? In order to verify the results for one structure or one wall it will be interesting to perform a kind of push-over curve (curve displacement-shear base response).

210 Maybe a table with the description of the building types from 1A to 6C would be useful for self-understanding of the paper accompanied by the building damage threshold for each typology (these thresholds are a very important issue, that can modify completely the results, and in my opinion this issue is not sufficiently treated in the paper).

We have rephrased the text explaining figure 4 (line 191ff) to make the interpretation of the curves clearer. The curves in figure 4 represents the structural limit of each building class. Depth-pressure combinations above the

215 line exerts forces greater than the building can withstand, however the damage (e.g. collapse vs. partial collapse vs. weakening/cracks) caused by these forces is still likely to be proportional to the magnitude of the excess forces. While we would like to extend this work into determining proportional losses, the (current) lack of data and observations limits us in this regard, this is discussed in the limitations and discussions section.

220 Line 195 and following: OK, but is this representative? Most hyperconcentrated flows in Arequipa carry boulders given the environment there, and the impact of those is very important although not taken into account at all in your study …

The simple answer here is yes, boulder induced damage is critical for determining lahar induced damage, and is noted throughout the manuscript (lines 56 – 65, 127 – 128, 383 – 390). However, boulder damage (at least, a

225 flows' boulder carrying capacity) is proportional to depth and dynamic pressure (*velocity*). In this study, we see that a large amount of damage will occur even without boulders, and that exposure (the proximity and orientation of houses relative to the quebrada) dominates. In this context, detailed studies of boulder carrying, sizes etc. are less important compared to quantifying (and trying to reduce) the exposure. The effect of boulders will be addressed thoroughly in a broader discussion of assumptions.

230

**Lahar rheology**

General remark: This is a long overview but it is not clear why you expose all this to introduce finally the quadratic rheology model. Rewrite this section illustrating from the beginning the quadratic rheology model and how it compares to other existent models and explain your choice rather than the general information which is rather

235 disconnected from the rest. Also, you are using a lot of technical vocabulary here concerning the rheology which

some part of the readers may not be familiar with. So, simplify and rather concentrate on providing definitions of critical terms such as non-Newtonian, Newtonian, etc.

This section has been greatly simplified and combined with the following section. Critical terms (non-Newtonian, Newtonian, etc.) are defined.

240

**Implementation in smoothed particle hydrodynamics**

Line 246: OK, but you need to describe at least a little bit what this model is and how it is different from classically used lahar modeling software such as Titan2D/3D, LaharZ, etc.

A brief comparison to common modelling approaches (such as Titan and LaharZ) will be added earlier to explain

245  why SPH is used in this application.

Line 257f: Again, I got lost here – what is your point?

The purpose of viscosity regularisation is to reduce computational cost, this is highlighted better in the revised manuscript. These technical details are crucial for readers who may want to reproduce the approach, but is kept

250  as brief as possible for readers not familiar with modelling/terminology.

Line 263: Where is m in the equation? How many validation simulations did you use and what data was chosen for it?

This is a typo, we have changed to parameter "$c$" and added more explanation on why it was set to 200 (validation

255  against analytical results).

**Lahar simulations**

Line 268: "may not represent any specific event…" So what is the use of it? This is not at its place here. You may have a general discussion section where such issues should be discussed, but at this point, the reader may not

260  think it is useful to continue the reading.

A discussion on flow scenarios (flow rate and rheology) will be placed in the Case Study section, along with a justification of why those flow rates were chosen. A clearer aim for the manuscript ("…investigate the effect of hazard (flow rate and rheology), exposure (building orientation) and vulnerability (building quality/type) components on building loss in Arequipa") also helps to address this comment.

265

Line 269: 12.5 cm resolution? At this resolution, the identified building blocks and houses should be much more accurate than what appears in Figure 2 and Figure 5!

The final terrain model resolution is lower than this (this detail was missing in the initial manuscript and will be added to the 'Case study' section).

270

Line 281: Again, remind what these stand for and why you can take the experimental values for Arequipa.

All symbols have now been clearly explained, discussion on rheology values are included in the Case study section (see previous)

275  Line 286: Hyperconcentrated flows can have high viscosities, no?

Correct, the viscosity of our hyperconcentrated flows were ~10 times higher than that of clear water (see table 2). We understand the confusion of this sentence and have reworded it to highlight that dispersive stresses are higher in our debris flows.

Line 289: 45 seconds: why? Computational time and resulting cost? Maybe optimize this paragraph.

Line 290f: I do not agree here. Lahar surges do not always have higher depths than a steady lahar flow. First, a steady lahar flow is not necessarily the opposite of a lahar surge but a different flow type/phase. Second, the surge can be quite small and there can be several small surges before the major lahar flow front arrives. It depends a lot on the location and the environmental conditions triggering the lahar. So careful here to not make it sound as a general rule.

This paragraph (line 289 to 291) is rewritten to address these comments - computational time (cost) limited the simulations to 45 seconds and therefore our scenarios most represent the damage caused by higher velocity and depth surges/waves in lahars than steady lahar flow.

**Flow behaviour**

Line 297ff: Not clear. What is your point here?

Line 300: This is not new…

Lines 294 – 302 are a brief description of the main flow features shown in Fig. 5 to help the reader in their interpretation and highlight aspects that are important for the following discussions. This paragraph has been modified to make the point of the paragraph clear.

Line 300ff: Decreasing velocity can also be due to friction effects and turbulence along the channel walls.

Different friction effects and turbulence are a result of the differences in rheology, this point has been added.

Line 308: "buildings oriented parallel to the channel": this depends how you define the long axis of the building. Looking at Figure 2, this is not very clear to me. What do you mean by "broad understanding of normal pressure"?

Our explanations of buildings, blocks and orientation to the channel are improved in the case study section. Bad grammar of "broad understanding of normal pressure" fixed.

Line 313: illustrate this with a graphic illustration. "Higher EW pressures …" Not sure I understand what you mean. Flow looses velocity and depth when spreading in side-roads?

Figure 5 has been enhanced to show directions of pressure.

Line 316: Pressure magnitude: you need to define this term.

Line 323: normal pressure: need to define this term!

In response to reviewer #2's comments, this has been refined to (1) add discussion to the section on lahar simulations explaining how dynamic pressure is calculated and (2) modify the flow behaviour section to explain the difference between using the velocity magnitude (scalar, not normal to walls) and normal velocity to determine forces exerted on the walls.

Line 329: you were mentioning earlier that you consider only initial lahar surges, so this is not really a conclusion of the modeling, is it?

Correct, this has been changed.

Line 330: Why is this a consequence of the short timeframe?

We have expanded an earlier (calculation of moments) part to explain that equalisation of hydrostatic pressure requires the fluid to enter the building and rise to the same height as the outside. This takes a reasonable amount of time, much less than the duration of the surges modelled here.

Line 334: "higher pressures": you mean higher dynamic pressure?

Correct – although changed to higher flux of momentum in response to reviewer #2's feedback.

Line 335:  If this indicates elevation differences, it would be helpful to have a map illustrating what elevation differences there are in this study area.

This wording has been refined, it refers to elevation of the cross-streets.

Line 336: acting on blocks: rather buildings?

Data from the simulation is obtained per-block (which is now explained in an earlier section), so we take care to describe the affects related to the block, rather than individual buildings.

Line 337ff: reformulate, this paragraph is not clear. Also, I am skeptical that higher density generally causes larger dynamic pressure …can you provide reference?

As pressure is proportional density time the square of velocity, for the same velocity, dynamic pressure will be larger. See Jenkins et al. (2015) as an example. We have reformulated the paragraph to describe the differences in dynamic pressures between rheologies.

Line 338: near perpendicular walls: and not near, what happens?

Added explanation.

Line 340: The dilatant component is lower for debris flows than for Newtonian and hyperconcentrated flows, so the latter decrease in velocity quite rapidly when diminishing in depth which also reduces the pressure for these flows…

The dilatant component is much higher for debris flows than hyperconcentrated and Newtonian flows – see table 2.

**Application of critical depth-pressure curves**

Line 342: "along the block" – means what?

These building, block and orientation descriptions are defined earlier in the manuscript.

355 Line 348f: Not clear, "well above the critical curves for each block". General remark: you switch between buildings and blocks and it is in the end not very clear for what you realize your modeling and how do you generate the curves for one block? Is it the mean value of all buildings contained in the block? In this case, we need to have more information on each block…

The role of the critical curves, difference between buildings and blocks are all explained in earlier sections to
360 make this part clearer.

Line 353: fluid height – be consistent in the use of your terms to make it easier for the reader to follow. Either use fluid height or flow depth.

Depth is now used throughout.
365

Line 354: Which means no damage to expect?

Not necessarily, this is explained in the expanded discussion section.

Line 357: lower depths? This is interesting: I would have thought that depth rises when flow impacts a
370 perpendicular wall. At least this is what you see frequently with … it would be interesting to compare your methodology with another location and see if you find similar relationships.

Depths are lower for a number of reasons: (1) perpendicular walls are further from the channel, (2) overbank flow is occurring, meaning that the flow is not directly impacting perpendicular walls, but spreading along cross-streets and (3) velocities are generally lower in these cross-streets, reducing the size of run-up on a wall, this will be
375 added.

Line 359: "effect of vulnerability": this means what?

This sentence is more suited to discussion and has been moved.

380 Line 371: explain in more detail what this means "reducing the size of the applied moment".

Explained in better detail; the applied moment is smaller as a result of the low depth.

Line 375: across scenarios: means that this can be observed in several scenarios? How many did you have in total? Other studies also indicate that distance from the channel plays a more important role than the building
385 type itself…

Added more detail on scenarios this was observed. Also added references to other studies that confirm these findings (that distance is more important than building type).

**Discussion**

390 Line 380ff: Reformulate, this paragraph is not clear.

Line 389: "applied to ultimate moment and damage through other actions" – how are these defined?

Line 390: comprehensive data: like what?

This paragraph has been reformulated and definitions have been clearly explained.

395      Line 393: Repetition, see line 329

This has been moved to the previous section, as a better explanation.

Line 395ff: depends on localization of the building respective to the channel: first row, second row, screening effect, etc.

400      Added more detail to explain this could be an effect.

Line 409: this is not new…

Added references.

405      Line 410: "may be too complicated" – no, not necessarily, it just needs better assessments, more data and detailed field studies.

This doesn't have much place in the manuscript, removed the sentence.

**Conclusion**

410      Line 416: not new

Added references.

Line 417ff: This paragraph comes somehow out of the blue and is quite disconnected from the previous rather technical descriptions. The recommendations for retrofitting are not enough put in relation with the results of

415      your modeling, it would maybe be a better option to add this into the discussion and link it more specifically to your results.

Added these elements to the discussion.

Line 425: here you say the approach can be generalized but given all the restrictions and constraints you have for

420      Arequipa, this sounds rather unlikely. The best way would be to propose a section where you apply this approach to a different study area and shortly explain how you validate your approach. Or you need to explain more in detail why and how this approach can be generalized.

The vulnerability calculations are necessarily specific to Arequipa, however the approach to quantifying losses (using bending moments, calculating pressures from numerical modelling) can be applied to other areas. This has

425      been expanded in the discussion section now.

Line 429: Large-scale indicators: such as?

Line 430: In my opinion, you are not modeling vulnerability. You use building type data, do lahar modeling and cross this information to generate building performance curves…

430      Line 431: "refine the indices in order to focus": only if data is available as for now you use a lot of data extrapolated from other sources.

The conclusion will be largely rewritten to reinforce the main 'novel' aspects of this work. These are valid comments, and have been taken into consideration when writing the conclusion.

435 **Figures**

Figure 1: need to complete the legend indicating what the yellow and orange lines stand for as well as the orange areas (built area: extent of the city at present?); reduce the size of the North arrow. The NE end of the quebradas Venezuela, Huarangal and Andamayo are quite low and appear disconnected from the foothill of the volcano – is there a reason for this presentation?

440 We have modified this figure and the caption to improve understanding. This is a simplified map to set the context for readers.

Figure 2: I think you need to better define what you understand by building block. Since you refer a lot to the works of Thouret et al. 2013 and 2014, his definition of building blocks seems to be different from what you

445 delineate here as a block. Clarify. Also, the delineation of your blocks is not easy to understand: for example in fig. 2B, bottom left, you distinguish three blocks dark green, light green and dark blue; on the opposite side of the road, the light violet block appears to be a single block although you might also have distinguished at least two; same for the light green block further North on the right side … if so is your choice, at least explain how you determine.

450 Additional explanation has been added to the Case study section of the manuscript to improve understanding of buildings and blocks.

Figure 3: title of legend "Design compressive stress" – means what? Is it the compressive stress? "Building Type" – no capital letter for type.

455 Added explanation of "Design compressive stress" to the caption (the compressive stress the building can withstand according to the standard). Modified Building type axis label.

Figure 4: Maybe simplify where possible this group of graphics: no need to put the "depth" legend title everywhere; reduce the appearing scale numbers (no need to detail every 0.25, just indicate 0,5; 1 and leave the

460 little markers for subunits). There is a lot of information presented, so simplify a maximum.
Simplified figure following these recommendations.

Figure 5: what do the arrows indicate on the lower left of each image? Images are quite small text is a little too big. Might be helpful to have an image without any modeling result where you locate the channel limits, outline

465 the buildings and indicate flow direction.
Figure 6: same remark as above; legend too big, no need to reproduce the same legend for all three image groups. The pressure value to the left is too isolated, difficult to understand what this means.
Made images larger, reduced size of the text and modified arrows (indicates North). The channel limits and buildings are outlined in Figure 2 (now modified slightly to explain this).

470

Figure 7: OK, very nice.
Figure 8: OK, very nice.

Figure 9, 10 and 11:  Difficult to distinguish the individual curves:  maybe fine lines of different colors would be
easier to read here.

Made lines finer to help distinguish curves.

Figure 12 and 13:  Legend title "loss fraction" is a little strange not associated with percentages. Flow rate needs
a unit! For the rest: same remark as above: simplify.

Simplified curves, modified loss fraction to percentages.

**Tables**

Table 1: Associate table with a figure where these numbers can be seen.

Association to figure 2c included in caption.

Table 2:  OK – I am not an expert in streamflow/lahar rheology, so I am not able to evaluate the correctness of
these parameters.  However, I understand these parameters are not your own modeling results but used to model,
so provide reference for the source of these values in this table.

References added to the table.

Jenkins, S., Phillips, J., Price, R., Feloy, K., Baxter, P., Hadmoko, D., and de Bélizal, E.:
Developing building-damage scales for lahars: application to Merapi volcano, Indonesia,
Bulletin of Volcanology, 77, 1-17, 10.1007/s00445-015-0961-8, 2015.
Zheng, K., Sun, Z.-c., Sun, J.-w., Zhang, Z.-m., Yang, G.-p., and Zhou, F.: Numerical
simulations of water wave dynamics based on SPH methods, Journal of Hydrodynamics, Ser.
B, 21, 843-850, Doi: 10.1016/s1001-6058(08)60221-8, 2009.

---

## Author Comment (AC2) · 24 Feb 2017

We would like to thank the reviewer for their suggestion and further suggested corrections. We incorrectly (in the manuscript, not in calculations), referred to dynamic pressure as $\rho v^2$ when we actually used $0.5\rho v^2$. As you have mentioned, the dynamic pressure is a scalar quantity and therefore using terminology as 'directional components of dynamic pressure' could be misleading. However, we disagree with the suggestion to adopt momentum flux instead.

Dynamic pressure is used extensively in literature examining the forces applied to buildings by fluids (see e.g. Roos (2003), Zeng et al. (2015), Zuccaro et al. (2008) and Jenkins et al. (2015)) and the theoretical basis for using dynamic pressure is sound, based on Bernoulli's theorem: when a fluid interacts with a fixed solid surface, the

velocity normal to the surface is zero and the total pressure (i.e. force per unit area) is given as the stagnation pressure (see e.g. Landau and Lifshitz, 1959 pp. 11-21)

$$P_{max} = P_{static} + 0.5\rho v^2$$

Where v is the velocity normal to the wall.

This was not explained in enough detail in the original manuscript, and discussion of 'directional components of pressure' is misleading. To solve this, we intend to (1) add discussion to the section on lahar simulations explaining how dynamic pressure is calculated and (2) modify the flow behaviour section to explain the difference between using the velocity magnitude (scalar, not normal to walls) and normal velocity to determine forces exerted on the walls.

Please also note the supplement to this comment:
http://www.nat-hazards-earth-syst-sci-discuss.net/nhess-2016-282/nhess-2016-282-AC2-supplement.pdf

**Supplement:**

**Quantifying lahar damage using numerical modelling**

Stuart R. Mead[1,2], Christina Magill[1], Vincent Lemiale[2], Jean-Claude Thouret[3] and Mahesh Prakash[2]

[1]Risk Frontiers, Department of Environmental Science, Macquarie University, Sydney, Australia
[2]Commonwealth Scientific and Industrial Research Organisation, Clayton 3168, Victoria, Australia
[3]Laboratoire Magmas et Volcans UMR6524 CNRS, IRD and OPGC, University Blaise Pascal, Campus Les Cézeaux, 63178 Aubière , France

*Correspondence to:* Stuart R. Mead (Stuart.Mead@mq.edu.au)

**Response to Reviewer #2:**

The authors identify that both hydrostatic and hydrodynamic forces contribute to the bending moment (and eventually the collapse) of the walls of a building. The former is related to the flow depth whereas the latter is related to its momentum. For describing the momentum transported by the lahar, the authors introduce the concept of "dynamic pressure", defined by the authors as $\rho v^2$ (see line 294), and describe the components of the force as "directional components of dynamic pressure" (see eg: line 305). Although, I agree with the adopted philosophy, I suggest to slightly modify the used nomenclature. In fact, the "dynamic pressure", is usually defined as ½ $\rho v^2$ and it is a scalar. It is used in the context of the Bernoulli theorem for describing the variations of the fluid pressure along a streamline. I propose that, the author adopt the concept of flux of momentum, which is a tensor defined as (see eg: Landau and Lifshitz, 1959):

$$\Pi = P\mathbf{n} + \rho\mathbf{v}(\mathbf{v} \cdot \mathbf{n})$$

where is the density of the fluid, $\mathbf{n}$ is the unitary vector normal to a surface, and $\mathbf{v}$ is the flow velocity. Considering a surface normal to the flow (parallel to *n*), the component of the flux of momentum perpendicular the surface becomes $P + \rho v^2$, whereas when the velocity of the fluid is parallel to the surface, the only component of the flux of momentum perpendicular to the surface is *P*. The flux of momentum is related to the force exerted by a wall to deviate or stop the flow.

We would like to thank the reviewer for their suggestion and further suggested corrections. We incorrectly (in the manuscript, not in calculations), referred to dynamic pressure as $\rho v^2$ when we actually used ½ $\rho v^2$. As you have mentioned, the dynamic pressure is a scalar quantity and therefore using terminology as 'directional components of dynamic pressure' could be misleading. However, we disagree with the suggestion to adopt momentum flux instead.

Dynamic pressure is used extensively in literature examining the forces applied to buildings by fluids (see e.g. Roos (2003), Zeng et al. (2015), Zuccaro et al. (2008) and Jenkins et al. (2015)) and the theoretical basis for using dynamic pressure is sound, based on Bernoulli's theorem: when a fluid interacts with a fixed solid surface, the velocity normal to the surface is zero and the total pressure (i.e. force per unit area) is given as the *stagnation pressure* (see e.g. Landau and Lifshitz, 1959 pp. 11-21)

$$P_{max} = P_{static} + 0.5\rho v^2$$

Where *v* is the velocity normal to the wall.

This was not explained in enough detail in the original manuscript, and discussion of 'directional components of pressure' is misleading. To solve this, we intend to (1) add discussion to the section on lahar simulations

explaining how dynamic pressure is calculated and (2) modify the flow behaviour section to explain the difference between using the velocity magnitude (scalar, not normal to walls) and normal velocity to determine forces exerted on the walls.

**Minor corrections**

In equation (1), *b* is referred as the thickness of the bricks used in the wall (see line 145). Later, at line 147, *b* is referred as the wall thickness. This is different for walls with a thickness greater that the width of a single brick. Do you consider only walls made of a single layer of bricks?

Here, we assume that brick and wall thickness are equal (i.e. only single layer walls), as observed in the field investigation – this has been clarified in the manuscript and *b* now consistently refers to wall thickness throughout the manuscript.

Line 149: In this context, I suggest to specify that the "normal forces" are vertical forces.

Thank you for the suggestion, this has been changed and helps in reducing confusion between terms.

Line 170: Symbol $a \rightarrow a_v$

Changed.

Line 190: "The critical height". Do you mean "The critical depth"?

Correct. Height vs. depth inconsistencies have been fixed throughout the manuscript, depth is used throughout.

Line 223: Usually the coefficient $\mu$ (eq.7) is defined "viscosity" only when the exponent *m* is equal to 1.

Commonly, the symbol *k* would be used, however that clashes with other symbols used in the building strength sections. We have changed the symbol to $k_\mu$ to reduce the confusion.

Line 252: "equation 3". Perhaps you mean eq. (7) or (8)?

Lines 256 and 262: "equation 4"? Perhaps eq.(7)? Please check.

References to equations are corrected.

Line 305: The "directional components of the dynamic pressure ...". Pressure, and dynamic pressure are scalars (see above).

See previous response.

Lines 315-323: Please check if the concepts of "normal stress" is more appropriate than "normal pressure component".

This part has been changed slightly in response to the "flux of momentum" suggestion. Normal stresses would differ by building and material properties, whereas the normal pressure (now flux of momentum) does not. In this context, it is more appropriate to use flux of momentum.

Line 437: contribution number "XXX", please provide the number if necessary.

Number provided.

80    Jenkins, S., Phillips, J., Price, R., Feloy, K., Baxter, P., Hadmoko, D., and de Bélizal, E.: Developing building-damage scales for lahars: application to Merapi volcano, Indonesia, Bulletin of Volcanology, 77, 1-17, 10.1007/s00445-015-0961-8, 2015.
Roos, W.: Damage to buildings, Delft Cluster, 2003.
Zeng, C., Cui, P., Su, Z., Lei, Y., and Chen, R.: Failure modes of reinforced concrete columns
85    of buildings under debris flow impact, Landslides, 12, 561-571, 10.1007/s10346-014-0490-0, 2015.
Zuccaro, G., Cacace, F., Spence, R. J. S., and Baxter, P. J.: Impact of explosive eruption scenarios at Vesuvius, Journal of Volcanology and Geothermal Research, 178, 416-453, 10.1016/j.jvolgeores.2008.01.005, 2008.
90

---

## Author Response (AR1)

**Quantifying lahar damage using numerical modelling: response and list of changes**

Stuart R. Mead, Christina Magill, Vincent Lemiale, Jean-Claude Thouret and Mahesh Prakash

Again, we would like to thank the reviewers and editors for their time taken in review of this manuscript. A broad response to reviewers' comments has already been provided (see http://www.nat-hazards-earth-syst-sci-discuss.net/nhess-2016-282/#discussion). This list of changes forms a point-by-point response to each reviewer with changes noted.

Each change is noted and tracked in the document that follows these responses. For convenience, we have summarised the major changes as the following:

- Moved detail on the calculation of bending moments and ultimate strength to appendices.
- Added detail (in appendices) on calculation of shear strength.
- Merged Lahar rheology section with Implementation in smoothed particle hydrodynamics section.
- Added subsections to discussion highlighting, critically evaluating and justifying assumptions.
- Improved figures 1-6 and 9-13 in response to reviewers comments.

Sincerely,

Stuart Mead

**Point-by-point response**

| Reviewer | Summary of reviewer's Comments or Requirements | Corrections/amendments made | Revised page/line reference |
|---|---|---|---|
| **General/title and abstract comments** | | | |
| Reviewer 1 | Building vulnerability: A general confusion is created by the use of building types from other studies (Thouret et Al., 2013, 2014) which are referred to as building vulnerabilities. Clearly define what you mean by vulnerability. | We are now referring to these as simplified structural classes. An additional table that provides a description of each building type and typology and their simplified structural class has been included. | NA |
| Reviewer 1 | Building vs block: it is not always clear throughout the paper what you are taking into account for your modeling and how data is aggregated. … There is more explanation needed here. | Building, block and block orientation will be comprehensively discussed in the Case study section, highlighting the relationship between them. Figure 2 (which will be modified) and Table 1 also help to explain the number of buildings per block, and composition. | NA |
| Reviewer 1 | Lahar rheology and modeling: A lot of detailed description is provided on lahar rheology with the result that the reader remains relatively confused facing a complex topic with lots of | We have reduced the amount of discussion on lahar rheology to focus only on the core details needed to understand this study. Additional explanation on SPH and why it was chosen over other approaches (also explained) is provided. | NA |

| | | | |
|---|---|---|---|
| | technical terms in a few paragraphs. It is not clear from the structure of the text, what objective this section serves. A reorganization of this section might help to outline more specifically why the SPH model has been chosen, what its differences are compared to more commonly used lahar modeling software and what interest the use of this model has in terms of creating the depth-pressure curves. | | |
| Reviewer 1 | Structural failure model similar to those employed by Roos (2003): Roos (2003) made his classifications and his study (comparison of the loads on the structure with the strength of the structures) for masonry buildings in the Netherlands, so the transposition for Arequipa buildings should be argued. | This is addressed in an expanded 'limitations and discussion' section and explanation of bending moment calculations. | NA |
| Reviewer 2 | Lines 315-323: Please check if the concepts of "normal stress" is more appropriate than "normal pressure component". Line 305: The "directional components of the dynamic pressure ...". Pressure, and dynamic pressure are scalars (see above) | We have modified text to ensure our consistency with established terminology through the manuscript. | NA |

| Reviewer 1 | Title Better: lahar induced damage | Title changed to 'Examining the impact of lahar on buildings using numerical modelling' | NA |
|---|---|---|---|
| Reviewer 1 | Line 15: What is the "relative importance" of lahar hazard, etc.? | 'Relative' as a word is irrelevant here, changed to "…examine the importance of lahar hazard..." | 16 |
| **Introduction** | | | |
| Reviewer 1 | Line 24/25: may add here Vallance (2000) as a reference | The reference to Vallance and Iverson (2015) is from the second edition of Encyclopaedia of Volcanoes were Vallance (2000) is from the first. They are functionally the same work. | 25 |
| Reviewer 1 | Line 28: again, careful with the word "uncertainty" here, I think it is not adequate | Changed to '…varying number of elements…" here and in the abstract. | 28 |
| Reviewer 1 | Line 34: clarify "relative" – is this from a statistical point of view? | See previous response, 'relative' is removed. | 34 |
| Reviewer 1 | Line 75: "to identify strategies that may reduce building loss" – these strategies appear to be mentioned only in a short paragraph in the conclusion which appears quite disconnected from the rest. If this really is one of your objectives, this needs more argumentation and | Rather than strategies to reduce loss, the aim is better described as investigating the role of hazard (flow rate and rheology), exposure (building orientation) and vulnerability (building quality/type) components on building loss in Arequipa. This has been corrected and is consistent throughout the manuscript. | 76ff |

| | | | |
|---|---|---|---|
| | be put in relation in a more concrete way with your modeling results. | | |
| | **Case study: Quebrada Dahlia, Arequipa, Peru** | | |
| Reviewer 1 | Line 86: watch the spelling – torrenteRAS, one R at the end | Fixed throughout manuscript. | 88 |
| Reviewer 1 | Line 93: cite also and in first place Martelli 2011 | Added per NHESS reference style (chronological). | 90 |
| Reviewer 1 | Line 109: "relative effects" – again, don't understand the use of the word relative here. Be more precise. | Again, 'relative' is irrelevant in this context. | 111 |
| Reviewer 1 | Line 118: I don't understand the use of "by" here before cross streets. Does this mean the cross streets and quebrada separate the buildings in a way that blocks are formed? | Modified figure 2a and reworded sentence to clearly explain how the buildings are separated into to five city blocks. | 121 |
| Reviewer 1 | Line 119: the "general approach" needs to be explained at least in a few sentences for those who do not have the time to read Thouret et al (2014). | The following sentences explained the building classification method of Thouret et al. (2014), this has been reworded to make it clear. | 123 |

| Developing building vulnerability relationships | | | |
|---|---|---|---|
| Reviewer 1 | Line 133 and 135: Need to give more detail on these models/approaches. | A brief description of the concept of these approaches has been provided. | 139-141 |
| Reviewer 1 | Line 134-135: this is a very important assumption "Stresses for buildings in Arequipa are calculated using the approach specified in Australian Standard (AS) 3700-2011" and some better explanations should be provided. Line 135-136: Reformulate the sentence; it is not clear what you mean here. Line 137: "While some specifications in the standard may not be relevant for Arequipa, the calculation method is still valid for the area provided construction material properties from Arequipa are used as inputs." Yes, some specifications in the standard are not relevant for Arequipa, and the calculation method are not just related to the material properties, but also to the construction mode and the behavior, so "the methods are still valid" is not necessarily obvious, but should be argue in this way. | These comments have been dealt with simultaneously. A better explanation of why the standard can be relevant for Arequipa is provided, and the effects of this assumption is mentioned here and discussed in the reworked discussion section. | 129-158 |

| | | | |
|---|---|---|---|
| Reviewer 1 | Also important simplification or assumption "In these models, masonry walls are presumed to fail when the applied bending moment and shear forces are greater than the calculated ultimate bending moment and shear force the walls can withstand. We only consider the maximum bending moment here as preliminary investigations suggested the force required to overcome the ultimate moment was consistently lower than the force required overcoming the ultimate shear force". This should be justified by literature investigation, experimental test or numerical simulation. The behavior of masonry wall is quite complex and can be (or generally is) a mix of flexural and shear behavior, in this case, considering just the maximum bending moment is an important assumption that can affect the global results and conclusions. See macroelement masonnery behaviour software TREMURI paper S. Lagomarsino, A. Penna, A. Galasco, S. Cattari, TREMURI program: an equivalent frame | The response to shear is important to consider, however the effect of this is minimal in our study. In an effort to simplify the manuscript we chose to leave discussion of shear out of the manuscript. However, for the reasons you identified, this was probably not the best idea. Instead, we have added additional information on the shear response in an appendix and made reference to it in this section. This way, the focus of the manuscript is maintained, but additional information is available to readers if needed. | 147 |

| | | | |
|---|---|---|---|
| | model for the nonlinear seismic analysis of masonry buildings, Eng Struct, 56 (2013), pp. 1787–1799.

Line 139: "preliminary investigations" – of what? Lab tests? Modeling? Field work? | | |
| Reviewer 1 | Moreover, when we use "design capacity specified in actual standards" for the constructions that have certainly "no design" or, in the better case "low code design", this affect the results and cannot be considered as representative for the studied area. | This has been partly addressed and discussed further in the limitations and discussion section. | 151ff and 350ff |
| Reviewer 2 | In equation (1), b is referred as the thickness of the bricks used in the wall (see line 145). Later, at line 147, b
is referred as the wall thickness. This is different for walls with a thickness greater that the width of a single brick.
Do you consider only walls made of a single layer of bricks? | Here, we assume that brick and wall thickness are equal (i.e. only single layer walls), as observed in the field investigation – this has been clarified in the manuscript and $b$ now consistently refers to wall thickness when appropriate. | 145 |
| Reviewer 1 | Line 146: "should not be greater than this value" – Why? Explain for those not familiar with this context. | Details on the calculation of bending moment has been moved to the Appendix. We have placed a short explanation in the Appendix that 0.2 MPa is the maximum tensile strength that can be assumed without | 448 |

| | | testing. This assumption is critical, and is explained and discussed in the limitations section. | |
|---|---|---|---|
| Reviewer 2 | Line 149: In this context, I suggest to specify that the "normal forces" are vertical forces. | Thank you for the suggestion, this has been changed in the appendix and helps in reducing confusion between terms. | 451 |
| Reviewer 1 | Line 150 and following: quite a lot of equations and parameters presented. Since you are basically using parameter values specified in the Australian Standard, it may not be absolutely necessary to detail all of these equations.
Line 169 and following: a little graphic illustration would be SO helpful here! | Details on calculation of ultimate moment (and now shear) have been added to the appendix as these components have been described previously (e.g. Zheng et al. (2009)) and are in standards. | 440ff |
| Reviewer 2 | Line 170: Symbol a→ $a_v$ | Changed | 471 |
| **Critical depth-pressure curves** | | | |
| Reviewer 1 | Line 186 and following: I am not able to follow here, there are too many classes from different sources, I got lost. Can you present this in an easier way than A0, 1A-2B etc.? Or at least provide an informative table describing what these mean? | A new table (Table 2) has been constructed relating building type, general description and structural class to help understanding of this and the previous sections. | 689 |

| Reviewer 2 | Line 190: "The critical height". Do you mean "The critical depth"? | Correct. Height vs. depth inconsistencies have been fixed throughout the manuscript, depth is used throughout. | 169 |
|---|---|---|---|
| Reviewer 1 | General remark: you are analyzing vulnerability of buildings so it is a little strange to use already defined vulnerability classes. It would sound more logical to refer to simple building types or categories for the Thouret et al classifications instead of calling these vulnerability classes. | We agree that 'vulnerability class' is a confusing term, we now refer to the classes as structural or simplified structural classes throughout the manuscript. | NZ |
| Reviewer 1 | For figure 4, you mean that the graphic represents the combination between the depth and the dynamic pressures for which you have failure of the building class? This could be assimilated as a limiting right under the line you do not have collapse and above the line you have collapse? In order to verify the results for one structure or one wall it will be interesting to perform a kind of push-over curve (curve displacement-shear base response). | We have slightly rephrased the text surrounding figure 4, discussing, as the reviewer states, that the curves indicate the structural limit of each class. Depth-pressure combinations above the line exerts forces greater than the building can withstand, however the damage (e.g. collapse vs. partial collapse vs. weakening/cracks) caused by these forces is still likely to be proportional to the magnitude of the excess forces. While we would like to extend this work into determining proportional losses, the (current) lack of data and observations limits us in this regard, this is discussed in the limitations and discussions section. | 170ff |
| Reviewer 1 | Maybe a table with the description of the building types from 1A to 6C would be useful | Table 2 (new) provides a description of each building type and structural class. To help with self-understanding, a shortened | Fig. 4 |

| | | | |
|---|---|---|---|
| | for self-understanding of the paper accompanied by the building damage threshold for each typology (these thresholds are a very important issue, that can modify completely the results, and in my opinion this issue is not sufficiently treated in the paper). | description for each class is added to figure 4. The extended discussion goes into further detail on the use and assumptions of damage thresholds. | |
| Reviewer 1 | Line 195 and following: OK, but is this representative? Most hyperconcentrated flows in Arequipa carry boulders given the environment there, and the impact of those is very important although not taken into account at all in your study … | The simple answer here is yes, boulder induced damage is critical for determining lahar induced damage. This is already noted throughout the manuscript (lines 56 – 65, 127 – 128, 383 – 390). However, boulder damage (at least, a flows' boulder carrying capacity) is proportional to depth and dynamic pressure (*velocity*). In this study, we see that a large amount of damage will occur even without boulders, and that exposure (the proximity and orientation of houses relative to the quebrada) dominates. In this context, detailed studies of boulder carrying, sizes etc. are less important compared to quantifying (and trying to reduce) the exposure. The effect of boulders on lahar damage is addressed in the discussion of assumptions. | |
| **Lahar rheology (*now 'Lahar rheology and implementation in smoothed particle hydrodynamics'*)** | | | |
| Reviewer 1 | General remark: This is a long overview but it is not clear why you expose all this to introduce finally the quadratic rheology model.  Rewrite | This section has been greatly simplified and combined with the following section. Critical terms (non-Newtonian, Newtonian, etc.) | 184-222 |

| | | | |
|---|---|---|---|
| | this section illustrating from the beginning the quadratic rheology model and how it compares to other existent models and explain your choice rather than the general information which is rather disconnected from the rest. Also, you are using a lot of technical vocabulary here concerning the rheology which some part of the readers may not be familiar with. So, simplify and rather concentrate on providing definitions of critical terms such as non-Newtonian, Newtonian, etc. | are defined, and an explanation of how this compares to other models has been provided. | |
| Reviewer 2 | Line 223: Usually the coefficient $\mu$ (eq.7) is defined "viscosity" only when the exponent $m$ is equal to 1. | This equation has been removed from the revised manuscript. | NA |
| Reviewer 1 | Line 246: OK, but you need to describe at least a little bit what this model is and how it is different from classically used lahar modeling software such as Titan2D/3D, LaharZ, etc. | A summary of the common modelling approaches (such as Titan and LaharZ) and why they are unsuitable for this study is provided, along with an explanation of why SPH is used in this application. | 196-200 |
| Reviewer 2 | Line 252: "equation 3". Perhaps you mean eq. (7) or (8)? | References to equations are corrected. | 209ff |

| | | | |
|---|---|---|---|
| | Lines 256 and 262: "equation 4"? Perhaps eq.(7)? Please check | | |
| Reviewer 1 | Line 257f: Again, I got lost here – what is your point? | The purpose of viscosity regularisation is to reduce computational cost; this is highlighted better in the revised manuscript. These technical details are crucial for readers who may want to reproduce the approach, but is kept as brief as possible for readers not familiar with modelling/terminology. | 214 |
| Reviewer 1 | Line 263: Where is m in the equation? How many validation simulations did you use and what data was chosen for it? | This is a typo, we have changed to parameter "$c$" and added more explanation on why it was set to 200 (validation against analytical results). | 219 |
| **Lahar simulations** | | | |
| Reviewer 1 | Line 268: "may not represent any specific event…" So what is the use of it? This is not at its place here. You may have a general discussion section where such issues should be discussed, but at this point, the reader may not think it is useful to continue the reading. | We have provided a better explanation of the choice of scenarios, and have highlighted the effect of these choices in the discussion section as appropriate. A clearer aim for the manuscript ("…investigate the effect of hazard (flow rate and rheology), exposure (building orientation) and vulnerability (building quality/type) components on building loss in Arequipa") also helps to address this comment. | 224-235 |

| | | | |
|---|---|---|---|
| Reviewer 1 | Line 269: 12.5 cm resolution? At this resolution, the identified building blocks and houses should be much more accurate than what appears in Figure 2 and Figure 5! | The final terrain model resolution is lower than this (this detail was missing in the initial manuscript and has been added to the 'Case study' section). SPH particle resolution is usually finer than terrain resolution, as in this case, to resolve important features of the flow. The resolution was chosen from a similar (preliminary) study in Mead et al. (2015) – referenced. | 119 |
| Reviewer 1 | Line 281: Again, remind what these stand for and why you can take the experimental values for Arequipa. | All symbols have now been clearly explained. | 238ff |
| Reviewer 1 | Line 286: Hyperconcentrated flows can have high viscosities, no? | Correct, the viscosity of our hyperconcentrated flows were ~10 times higher than that of clear water (see table 3). We understand the confusion of this sentence and have reworded it to highlight that dispersive stresses are higher in our debris flows. | 246-248 |
| Reviewer 1 | Line 289: 45 seconds: why? Computational time and resulting cost? Maybe optimize this paragraph. | This paragraph (line 289 to 291) is rewritten to address these comments - computational time (cost) limited the simulations to 45 seconds and therefore our scenarios most represent the damage | 249-252 |

| | | | |
|---|---|---|---|
| | Line 290f: I do not agree here. Lahar surges do not always have higher depths than a steady lahar flow. First, a steady lahar flow is not necessarily the opposite of a lahar surge but a different flow type/phase. Second, the surge can be quite small and there can be several small surges before the major lahar flow front arrives. It depends a lot on the location and the environmental conditions triggering the lahar. So careful here to not make it sound as a general rule. | caused by higher velocity and depth surges/waves in lahars than steady lahar flow. | |
| **Flow behaviour** | | | |
| Reviewer 1 | Line 297ff: Not clear. What is your point here? Line 300: This is not new… | Lines 294 – 302 are a brief description of the main flow features shown in Fig. 5 to help the reader in their interpretation and highlight aspects that are important for the following discussions – so is not necessarily highlighting things that are 'new', but making observations clear. This paragraph has been reworded (including removing the unclear sentence on L297). | 254-262 |
| Reviewer 1 | Line 300ff: Decreasing velocity can also be due to friction effects and turbulence along the channel walls. | Different friction effects and turbulence are a result of the differences in rheology, this point has been added. | 259 |

| Reviewer 1 | Line 308: "buildings oriented parallel to the channel": this depends how you define the long axis of the building. Looking at Figure 2, this is not very clear to me. What do you mean by "broad understanding of normal pressure"? | Our explanations of buildings, blocks and orientation to the channel are improved in the case study section. We have changed the sentences surrounding L308 to make it clear we are talking about orientation of walls (not buildings).

Bad grammar of "broad understanding of normal pressure" fixed ("…initial"). | 265-278 |
|---|---|---|---|
| Reviewer 1 | Line 313: illustrate this with a graphic illustration. "Higher EW pressures …" Not sure I understand what you mean. Flow looses velocity and depth when spreading in side-roads? | The descriptions around pressure actions have been modified. We believe this may help the reader understand the reasoning better than a figure might. | 270-275 |
| Reviewer 1 | Line 316: Pressure magnitude: you need to define this term.
Line 323: normal pressure: need to define this term! | In response to reviewer #2's comments, this has been refined by modifying the flow behaviour section to add discussion on the calculation of dynamic pressure is calculated and explain the difference between using the velocity magnitude (scalar, not normal to walls) and normal velocity to determine forces exerted on the walls. | 279-285 |

| Reviewer 1 | Line 329: you were mentioning earlier that you consider only initial lahar surges, so this is not really a conclusion of the modeling, is it? | Correct, this has been removed. | 296 |
|---|---|---|---|
| Reviewer 1 | Line 330: Why is this a consequence of the short timeframe? | We have expanded an earlier (critical depth-pressure curves) part to explain that equalisation of hydrostatic pressure requires the fluid to enter the building and rise to the same height as the outside. This takes a reasonable amount of time, much less than the duration of the surges modelled here. | 296-298 |
| Reviewer 1 | Line 334: "higher pressures": you mean higher dynamic pressure? | Correct – changed. | 301 |
| Reviewer 1 | Line 335: If this indicates elevation differences, it would be helpful to have a map illustrating what elevation differences there are in this study area. | This wording has been refined, it refers to elevation of the cross-streets. | 302ff |
| Reviewer 1 | Line 336: acting on blocks: rather buildings? | This paragraph has been reformulated to refer to walls. Data from the simulation is obtained per-block (which is now explained in an earlier paragraph), so we take care to describe the affects related to the block, rather than individual buildings. | 304-309 |
| Reviewer 1 | Line 337ff: reformulate, this paragraph is not clear. Also, I am skeptical that higher density generally causes larger dynamic pressure …can you provide reference? | As pressure is proportional density time the square of velocity, for the same velocity, dynamic pressure will be larger. See Jenkins et al. (2015) as an example. We have reformulated the paragraph to describe the differences in dynamic pressures between rheologies. | 306 |

| | | | |
|---|---|---|---|
| Reviewer 1 | Line 338: near perpendicular walls: and not near, what happens? | The difference between pressures applied to perpendicular and parallel walls has now been explained in this paragraph. | 304-309 |
| Reviewer 1 | Line 340: The dilatant component is lower for debris flows than for Newtonian and hyperconcentrated flows, so the latter decrease in velocity quite rapidly when diminishing in depth which also reduces the pressure for these flows… | The dilatant component is much higher for debris flows than hyperconcentrated and Newtonian flows – see table 2. The reformulated paragraph relating pressure differences to rheology should help to limit confusion on the effects of rheology on dynamic pressures. | 304-309 |
| **Application of critical depth-pressure curves** | | | |
| Reviewer 1 | Line 342: "along the block" – means what? | Changed to "…pressure acting on block walls…". Building, block and orientation descriptions are defined earlier in the manuscript. Explanation of how pressures are obtained also clarifies this meaning. | 311 |
| Reviewer 1 | Line 348f: Not clear, "well above the critical curves for each block". General remark: you switch between buildings and blocks and it is in the end not very clear for what you realize your modeling and how do you generate the curves | The role of the critical curves, difference between buildings and blocks are all explained in earlier sections to make this part clearer. | Previous. |

| | | | |
|---|---|---|---|
| | for one block? Is it the mean value of all buildings contained in the block? In this case, we need to have more information on each block… | | |
| Reviewer 1 | Line 353: fluid height – be consistent in the use of your terms to make it easier for the reader to follow. Either use fluid height or flow depth. | Depth is now used throughout. | 322 |
| Reviewer 1 | Line 354: Which means no damage to expect? | Not necessarily, this is explained in the expanded discussion section. | 350ff |
| Reviewer 1 | Line 357: lower depths? This is interesting: I would have thought that depth rises when flow impacts a perpendicular wall. At least this is what you see frequently with … it would be interesting to compare your methodology with another location and see if you find similar relationships. | Depths are lower for a number of reasons: (1) perpendicular walls are further from the channel, (2) overbank flow is occurring, meaning that the flow is not directly impacting perpendicular walls, but spreading along cross-streets and (3) velocities are generally lower in these cross-streets, reducing the size of run-up on a wall. | 324-328 |
| Reviewer 1 | Line 359: "effect of vulnerability": this means what? | This part of the sentence is more suited to discussion and has been moved. | NA |

| | | | |
|---|---|---|---|
| Reviewer 1 | Line 371: explain in more detail what this means "reducing the size of the applied moment". | Explained in better detail; the applied moment is smaller as a result of the low depth. | 340 |
| Reviewer 1 | Line 375: across scenarios: means that this can be observed in several scenarios? How many did you have in total? Other studies also indicate that distance from the channel plays a more important role than the building type itself… | Rephrased to say "…for all scenarios". Also added reference to other studies that confirm these findings (that distance is more important than building type). | 344 |
| **Discussion (now *Limitations and discussion*)** | | | |
| Reviewer 1 | Line 380ff: Reformulate, this paragraph is not clear.
Line 389: "applied to ultimate moment and damage through other actions" – how are these defined?
Line 390: comprehensive data: like what? | This paragraph has been reformulated and definitions have been clearly explained. | 383-393 |
| Reviewer 1 | Line 393: Repetition, see line 329 | This has been moved to the previous section, as a better explanation. | Previous |

| Reviewer 1 | Line 395ff: depends on localization of the building respective to the channel: first row, second row, screening effect, etc. | Added more detail to explain this could be an effect. | 394ff |
|---|---|---|---|
| Reviewer 1 | Line 409: this is not new… | This doesn't have much place in the manuscript, removed the sentence. | NA |
| **Conclusion** | | | |
| Reviewer 1 | Line 416: not new | Conclusion has been changed to make this not relevant | 425ff |
| Reviewer 1 | Line 417ff: This paragraph comes somehow out of the blue and is quite disconnected from the previous rather technical descriptions. The recommendations for retrofitting are not enough put in relation with the results of your modeling, it would maybe be a better option to add this into the discussion and link it more specifically to your results. | Added these elements to the discussion. | 415-424 |
| Reviewer 1 | Line 425: here you say the approach can be generalized but given all the restrictions and constraints you have for Arequipa, this sounds rather unlikely. The best way would be to propose a section where you apply this | The vulnerability calculations are necessarily specific to Arequipa, however the approach to quantifying losses (using bending moments, calculating pressures from numerical modelling) can be applied to other areas. This has been expanded in the conclusion. | 425-431 |

| | | | |
|---|---|---|---|
| | approach to a different study area and shortly explain how you validate your approach. Or you need to explain more in detail why and how this approach can be generalized. | | |
| Reviewer 1 | Line 429: Large-scale indicators: such as? Line 430: In my opinion, you are not modeling vulnerability. You use building type data, do lahar modeling and cross this information to generate building performance curves… Line 431: "refine the indices in order to focus": only if data is available as for now you use a lot of data extrapolated from other sources. | The conclusion will be largely rewritten to reinforce the main 'novel' aspects of this work. These are valid comments, and have been taken into consideration when writing the conclusion. | 425-438 |
| **Figures and tables** | | | |
| Reviewer 1 | Figure 1:  need to complete the legend indicating what the yellow and orange lines stand for as well as the orange areas (built area: extent of the city at present?); reduce the size of the North arrow.  The NE end of the quebradas Venezuela, Huarangal and Andamayo are quite low and appear disconnected from the foothill of the volcano – is | We have modified this figure and the caption to improve understanding. This is a simplified map to set the context for readers. | |

| | | | |
|---|---|---|---|
| | there a reason for this presentation? | | |
| Reviewer 1 | Figure 2: I think you need to better define what you understand by building block. Since you refer a lot to the works of Thouret et al. 2013 and 2014, his definition of building blocks seems to be different from what you delineate here as a block. Clarify. Also, the delineation of your blocks is not easy to understand: for example in fig. 2B, bottom left, you distinguish three blocks dark green, light green and dark blue; on the opposite side of the road, the light violet block appears to be a single block although you might also have distinguished at least two; same for the light green block further North on the right side … if so is your choice, at least explain how you determine. | Additional explanation has been added to the Case study section of the manuscript to improve understanding of buildings and blocks. | |
| Reviewer 1 | Figure 3: title of legend "Design compressive stress" – means what? Is it the compressive stress? "Building Type" – no capital letter for type. | Added explanation of "Design compressive stress" to the caption (the compressive stress the building can withstand according to the standard). Modified Building type axis label. | |

| | | | |
|---|---|---|---|
| Reviewer 1 | Figure 4: Maybe simplify where possible this group of graphics: no need to put the "depth" legend title everywhere; reduce the appearing scale numbers (no need to detail every 0.25, just indicate 0,5; 1 and leave the little markers for subunits). There is a lot of information presented, so simplify a maximum. | Simplified figure following these recommendations. | |
| Reviewer 1 | Figure 5: what do the arrows indicate on the lower left of each image? Images are quite small text is a little too big. Might be helpful to have an image without any modeling result where you locate the channel limits, outline the buildings and indicate flow direction.
Figure 6: same remark as above; legend too big, no need to reproduce the same legend for all three image groups. The pressure value to the left is too isolated, difficult to understand what this means. | Made images larger, reduced size of the text and modified arrows (indicates North). The channel limits and buildings are outlined in Figure 2 (now modified slightly to explain this). | |
| Reviewer 1 | Figure 9, 10 and 11: Difficult to distinguish the individual curves: maybe fine lines of different colors would be easier to read here. | Made lines finer to help distinguish curves. | |

| | | | |
|---|---|---|---|
| Reviewer 1 | Figure 12 and 13: Legend title "loss fraction" is a little strange not associated with percentages. Flow rate needs a unit! For the rest: same remark as above: simplify. | Simplified plots, modified loss fraction to percentages. | |
| Reviewer 1 | Table 1: Associate table with a figure where these numbers can be seen. | Association to figure 2c included in caption. | |
| Reviewer 1 | Table 2: OK – I am not an expert in streamflow/lahar rheology, so I am not able to evaluate the correctness of these parameters. However, I understand these parameters are not your own modeling results but used to model, so provide reference for the source of these values in this table. | References added to the table. | |

[revised manuscript text omitted]

**Figures**

[Figure]

[Figure]

**Figure 1. Location of Arequipa in relation to El Misti volcano, showing the main quebradas and the location of the Quebrada Dahlia study area.**

780

[Figure]

[Figure]

785

**Figure 2. Overview of Quebrada Dahlia study area, Arequipa, (a) aerial image with black outline showing study area, dashed outline showing channel banks and transparent lines showing streets in the area, (b) photogrammetric reconstruction of the surface and (c) individual buildings and building blocks identified from building surveys.**

[Figure]

**Figure 3. Range of design compressive stress for building types 1A - 6C defined in Thouret et al. (2014). Compressive**

stress capacity was calculated for every configuration of compressive strength ($f_c$), bedded area ($A_b$), and thickness coefficient ($k_t$) at brick widths ($b$) of 150 mm and 250 mm.

795

[Figure]

[Figure]

**Figure 4. Critical depth and dynamic pressures for failure of  structural classes A0, A and B for brick widths of 0.15 m (top) and 0.25 m (bottom). Shading of the lines indicate flow type and density, dotted lines and dashed lines represent the minimum and maximum forces required. Densities are for a Newtonian flow (NF, $\rho$ = 1000 kg·m$^{-3}$), hyper-concentrated flow (HCF, $\rho$ = 1500 kg·m$^{-3}$) and debris flow (DF, $\rho$ = 1915 kg·m$^{-3}$).**

800

[Figure]

[Figure]

Figure 5. Evolution of dynamic pressure and velocity magnitudes for a 75 m³s⁻¹ flow along Quebrada Dahlia for a Newtonian flow (NF), hyperconcentrated flow (HCF) and debris flow (DF). Arrows indicate North (green) and East (red) direction.

805

[Figure]

810

[Figure]

**Figure 6.** Directional components of dynamic pressure for a 75 m$^3$s$^{-1}$ flow along Quebrada Dahlia for a Newtonian flow (NF), hyperconcentrated flow (HCF) and debris flow (DF). **Maximum pressure is 25 kPa for magnitude and N-S pressures, 10 kPa for E-W pressure.**

[Figure]

**Figure 7. Comparison of mean pressure magnitude (grey lines) and mean normal pressure (black lines) on block 'West 2' in the parallel and perpendicular orientations for a 75 m³s⁻¹ flow along Quebrada Dahlia.**

[Figure]

820

**Figure 8. Mean normal pressures applied to each city block in the perpendicular and parallel orientations for a 75 m³s⁻¹ flow.**

[Figure]

[Figure]

**Figure 9. Critical depth-pressure curves for building classes A0, A and B subjected to Newtonian flow. Peak normal pressures and corresponding depths applied to each city block are plotted as points for each flow rate.**

[Figure]

830

[Figure]

**Figure 10. Critical depth-pressure curves for building classes A0, A and B subjected to a hyperconcentrated flow. Peak normal pressures and corresponding depths applied to each city block are plotted as points for each flow rate.**

[Figure]

[Figure]

**Figure 11. Critical depth-pressure curves for building classes A0, A and B subjected to a debris flow. Peak normal pressures and corresponding depths applied to each city block are plotted as points for each flow rate.**

[Figure]

840

**Figure 12. Building loss fraction for all flow scenarios where buildings assumed to have a brick width of 0.15 m.**

[Figure]

**Figure 13. Building loss fraction for all flow scenarios where buildings assumed to have a brick width of 0.25 m.**

845

**Tables**

**Table 1. Individual building type and vulnerability classes for each block in the Quebrada Dahlia study area. Block ID increases from north to south.**

| Block | ID | Type | Structural class | Structural class | Type | ID | Block |
|-------|----|------|------------------|------------------|------|----|-------|
| West 1 | 1 | 1A | A | A0 | 2A | 1 | East 1 |
|        | 2 | 1A | A0 | A0 | 2A | 2 | East 1 |
|        | 3 | 4 | B | A | 3 | 3 |  |
|        | 4 | 4 | A0 | A0 | 1B | 1 | East 2 |
|        | 5 | 2B | A0 | A | 3 | 2 |  |
|        | 6 | 3 | A | A0 | 1B | 3 |  |
|        | 7 | 4 | B | A0 | 1A | 4 |  |
| West 2 | 1 | 4 | B | B | 4 | 1 | East 3 |
|        | 2 | 4 | B | A | 3 | 2 |  |
|        | 3 | 4 | B | A | 3 | 3 |  |
|        | 4 | 4 | B |  |  |  |  |
|        | 5 | 3 | A |  |  |  |  |

850 **Table 2. Building types and simplified structural classes, from Thouret et al. (2014) (Thouret).**

| Typology | Building description | Simplified structural class |
|----------|----------------------|------------------------------|
| 1A | Unreinforced masonry of lapilli, ignimbrite or terra–cotta with no roof support structure (i.e. metal sheet roof) | A0 |
| 1B |  | A0 |
| 2A |  | A0 |
| 2B |  | A0 |
| 3 | Terra-cotta masonry with reinforced concrete roof. | A |
| 4 | Terra-cotta masonry with reinforced concrete frame and roof. | B |
| 5 | Historical ignimbrite building with mortar. | A |
| 6A | Ignimbrite masonry with reinforced concrete elements or modifications. | B |
| 6B |  | B |
| 6C |  | B |

**Table 3. Density, particle concentration and rheology coefficients for hyperconcentrated streamflow and debris flow simulations, taken from Govier et al. (1957); Julien and Lan (1991).**

| Flow type | Density (kg m$^{-3}$) | Particle concentration by volume (%) | Yield strength ($\tau_y$, Pa) | Viscosity ($\mu$, Pa s) | Dispersive stress coefficient ($\alpha$) |
|-----------|----------------------|--------------------------------------|-------------------------------|-------------------------|-------------------------------------------|
| Hyperconcentrated streamflow | 1500 | 30.3 | 0.94 | 0.0137 | $1.28 \times 10^{-5}$ |
| Fine-grained, matrix supported debris flow | 1915 | 55.5 | 0.672 | 0.0485 | 0.00224 |

855